# Fourier-based three-dimensional multistage transformer for aberration correction in multicellular specimens

**Thayer Alshaabi** [1,2] ✉, **Daniel E. Milkie** [1], **Gaoxiang Liu**[2], **Cyna Shirazinejad**[2], **Jason L. Hong** [2], **Kemal Achour**[2], **Frederik Görlitz** [2], **Ana Milunovic-Jevtic** [2], **Cat Simmons**[2], **Ibrahim S. Abuzahriyeh**[2], **Erin Hong**[2], **Samara Erin Williams**[2], **Nathanael Harrison**[2], **Evan Huang**[2], **Eun Seok Bae**[2], **Alison N. Killilea** [2], **Ian A. Swinburne**[2], **David G. Drubin**[2], **Srigokul Upadhyayula** [2,3,4] ✉ & **Eric Betzig** [1,2,5,6] ✉

High-resolution tissue imaging is often compromised by sample-induced optical aberrations that degrade resolution and contrast. Although wavefront sensor-based adaptive optics (AO) can measure these aberrations, such hardware solutions are typically complex, expensive to implement and slow when serially mapping spatially varying aberrations across large fields of view. Here we introduce AOViFT (adaptive optical vision Fourier transformer)—a machine learning-based aberration sensing framework built around a three-dimensional multistage vision transformer that operates on Fourier domain embeddings. AOViFT infers aberrations and restores diffraction-limited performance in puncta-labeled specimens with substantially reduced computational cost, training time and memory footprint compared to conventional architectures or real-space networks. We validated AOViFT on live gene-edited zebrafish embryos, demonstrating its ability to correct spatially varying aberrations using either a deformable mirror or postacquisition deconvolution. By eliminating the need for the guide star and wavefront sensing hardware and simplifying the experimental workflow, AOViFT lowers technical barriers for high-resolution volumetric microscopy across diverse biological samples.

As we peer deeper into living organisms to reveal their inner workings, our view is increasingly compromised by sample-induced optical aberrations. Numerous AO methods exist to compensate for these by using a wavefront shaping device that responds to a measurement of sample-induced aberration[1]. These methods differ in their complexity, generality, robustness and practicality. In our laboratory, dependable success was had using a Shack–Hartmann (SH) sensor to measure the aberrations imparted on a guide star (GS) created by two-photon excitation (TPE) fluorescence within the specimen[2], and we have used this approach extensively in adaptive optical (AO) lattice light sheet (LLS) microscopy (AO-LLSM) to study four-dimensional (4D) subcellular dynamic processes within the native environment of whole multicellular organisms[3].

Several recent approaches dispense with the cost and complexity of hardware-based wavefront measurement in favor of directly inferring aberrations from the microscope images themselves through

**Fig. 1 | AOVIFT workflow. a**, AOVIFT correction. An aberrated 3D volume is preprocessed and cast into a Fourier embedding, which is passed to a 3D vision transformer model to predict the detection wavefront. A DM compensates for this aberration, enabling acquisition of a corrected volume ($D$, depth; $H$, height; $W$, width). **b**, The Fourier embedding, $\mathcal{E}$. The Fourier transform of the 3D volume is embedded into a lower space ($\mathcal{E} \in \mathbb{R}^{\ell \times d \times d}$), consisting of three amplitude planes ($\alpha_1, \alpha_2, \alpha_3$) and three phase planes ($\varphi_1, \varphi_2, \varphi_3$), each of size $d \times d$ where $d$ is the Fourier embedding size. **c**, AOVIFT model. The Fourier embedding is input to

a dual-stage 3D vision transformer model. At each stage, the $\ell$ Fourier planes are tiled into $k$ patches (Patchify), applying a radially encoded positional embedding to each patch. These patches are passed through $n$ Transformer layers. At the end of each stage, a residual connection is added, and the patches are merged back to the shape matching the stage input (Merge patches). After all stages, the resulting patches are pooled (GlobalAvgPool) and connected with a dense layer to output the $z$ Zernike coefficients.

machine learning (ML)[4–8] (Supplementary Table 1). Based on our experience with a variety of specimens, any ML-AO approach suitable for AO-LLSM must meet the following specifications:

(1) Speed: to maximize the range of spatiotemporal events that can be visualized, the time for the ML model to infer the aberrations across any volume should be less than the time needed to image it—typically a few seconds in LLSM for a volume that encompasses a handful of cells.

(2) Robustness: the model must accurately predict the vast majority of aberrations encountered in practice—for AO-LLSM in zebrafish embryos, typically up to $5\lambda$ peak-to-valley (P–V), where $\lambda$ is the free-space wavelength, in any combination of the first 15 Zernike modes ($Z_0^0$ through $Z_4^{\pm4}$ Supplementary Fig. 1).

(3) Accuracy: the method should be able to recover close to the theoretical three-dimensional (3D) resolution limits of the microscope, regardless of the distribution of spatial frequencies within the specimen.

(4) Noninvasiveness: the method should provide accurate correction without unduly depleting the fluorescence photon budget within the specimen or perturbing its native physiology.

As none of the aforementioned ML-AO methods meet all these specifications, we endeavored to create one better suited to the needs of AO-LLSM. Our baseline model architecture, selected from an ablation study (Supplementary Note A, Supplementary Figs. 2–7), contains two transformer stages with patches of 32 and 16 pixels, respectively (Fig. 1c).

Priors can greatly improve the performance of any ML approach. For our method, we depend on the prior that each isoplanatic

subvolume (that is, having the same aberration) within the larger volume of interest contains one or more fluorescent puncta of true subdiffractive size. Here we introduce these by using genome-edited specimens expressing fluorescent protein-fused versions of AP2—an adapter protein that targets clathrin-coated pits (CCPs) ubiquitously present at CCPs located on the plasma membrane of all cells (Methods). While this entails a one-time upfront cost for each specimen type, it noninvasively produces a robust signal for AO correction that does not preclude simultaneously imaging another subcellular target that occupies the same fluorescence channel, provided they are computationally separable[9].

## Results

### Benchmark comparisons of AOVIFT to other architectures

We created five variants of AOVIFT by varying the numbers of layers and heads in each stage (Supplementary Table 2) to explore the tradeoffs between model size (number of parameters and memory footprint), speed (floating-point operations (FLOPs) required, training time and latency) and prediction accuracy (Supplementary Fig. 3). To compare these to existing state-of-the-art architectures, we developed 3D versions of ViT and ConvNeXt for AO inference in three and four different size variants, respectively (Supplementary Note B). We trained all models with the same set of $2 \times 10^6$ synthetic image volumes chosen to capture the full diversity of aberrations and imaging conditions likely to be encountered in AO-LLSM (Methods) and tested AOVIFT on a separate set of $10^5$ image volumes created to find the limits of its accuracy when presented with an even larger range of aberration magnitudes, signal-to-noise ratio (SNR) and number of fluorescent puncta (Supplementary Note C). We also tested the performance of

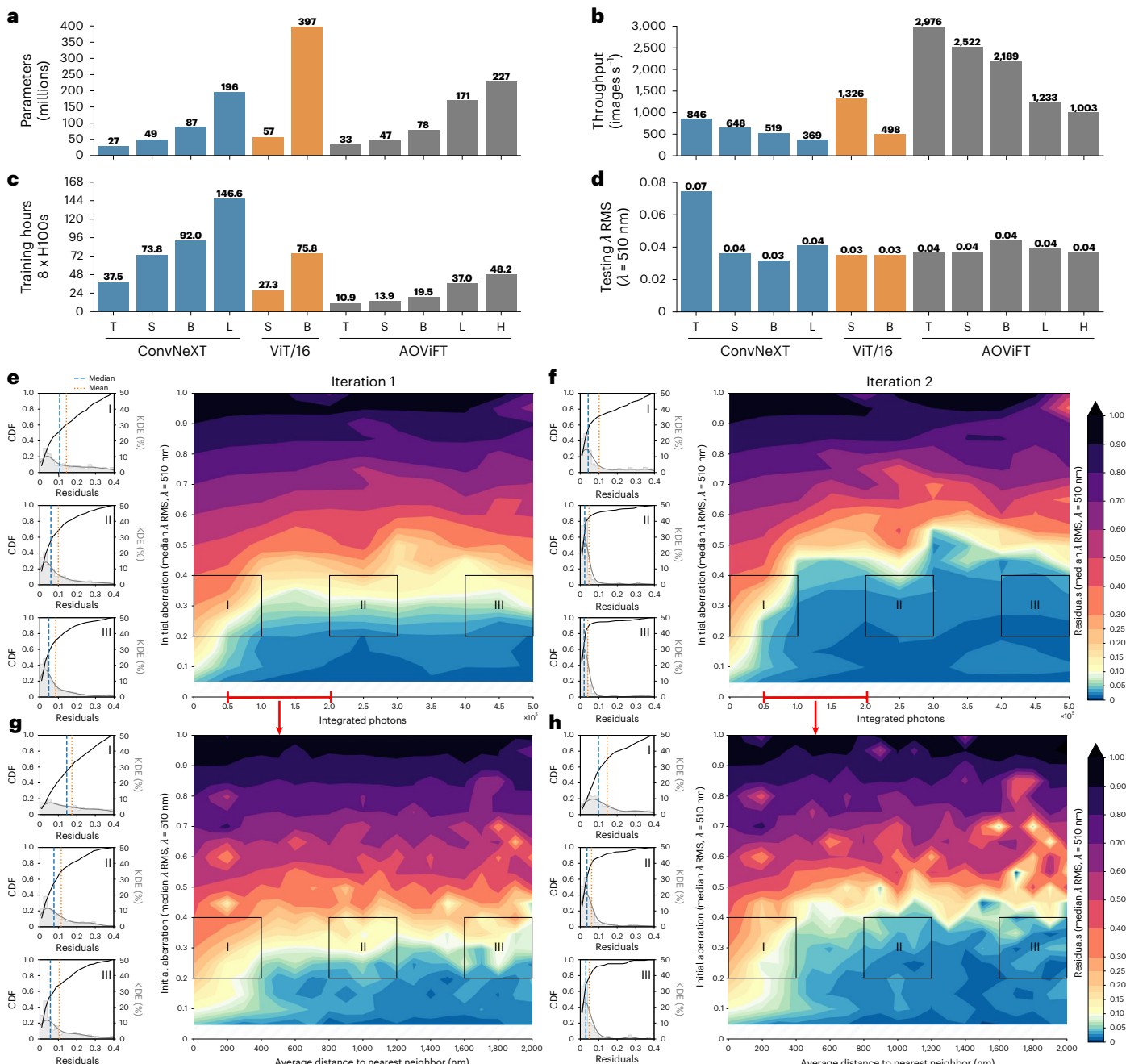

**Fig. 2 | Comparison of different state-of-the-art architectures when applied to 3D aberration sensing. a**–**d**, Comparison of model variants ConvNeXt-T/S/B/L (blue), ViT/16-S/B (orange) and AOViFT-T/S/B/L/H (gray). **a**, Total number of trainable parameters. **b**, Maximum predictions per second, using a batch size of 1,024 on a single A100 GPU. Higher values are better. **c**, Training time on eight H100 GPUs. **d**, Median λ RMS residuals over 10,000 test samples after one correction, with aberrations ranging between 0.2λ and 0.4λ, simulated with 50,000 to 200,000 integrated photons. **e,f**, Median λ RMS residuals using our Small variant of AOViFT model for a single bead over a wide range of SNR. **g,h**, Median λ RMS residuals using our Small variant of AOViFT model for several beads (up to 150 beads), simulated at photon levels from 50,000 to 200,000 per bead. Lower values are better for all performance indicators listed here, except for **b**. CDF, cumulative distribution function; KDE, kernel density estimation.

all models and variants on $10^4$ image volumes from a test set that contained only a single punctum in each (Fig. 2, Supplementary Figs. 8–9 and Supplementary Table 3).

Although all models but the smallest variant of ConvNeXt were able to reduce the median residual error in a single iteration of aberration prediction to less than the diffraction limit (Fig. 2d), AOViFT excelled in its parsimonious use of compute resources: training time using a node with eight NVIDIA H100 GPUs (Fig. 2c), training FLOPs (Supplementary Fig. 8c) and memory footprint (Supplementary Fig. 8f). This reflects the benefits of our multistage architecture: faster

convergence by learning features across different scales, accurate prediction even at comparatively modest model size (Fig. 2a), highest inference rate among the models tested (Fig. 2b) and fastest single-shot inference time ('latency'; Supplementary Fig. 8h). Given its small size and low latency, we chose the Small variant of AOViFT as our primary model for evaluation.

### In silico evaluations of AOViFT

Diffraction-limited performance is defined conventionally by wavefront distortions below ≈0.075λ root mean square (RMS) or λ/4 peak-to-valley,

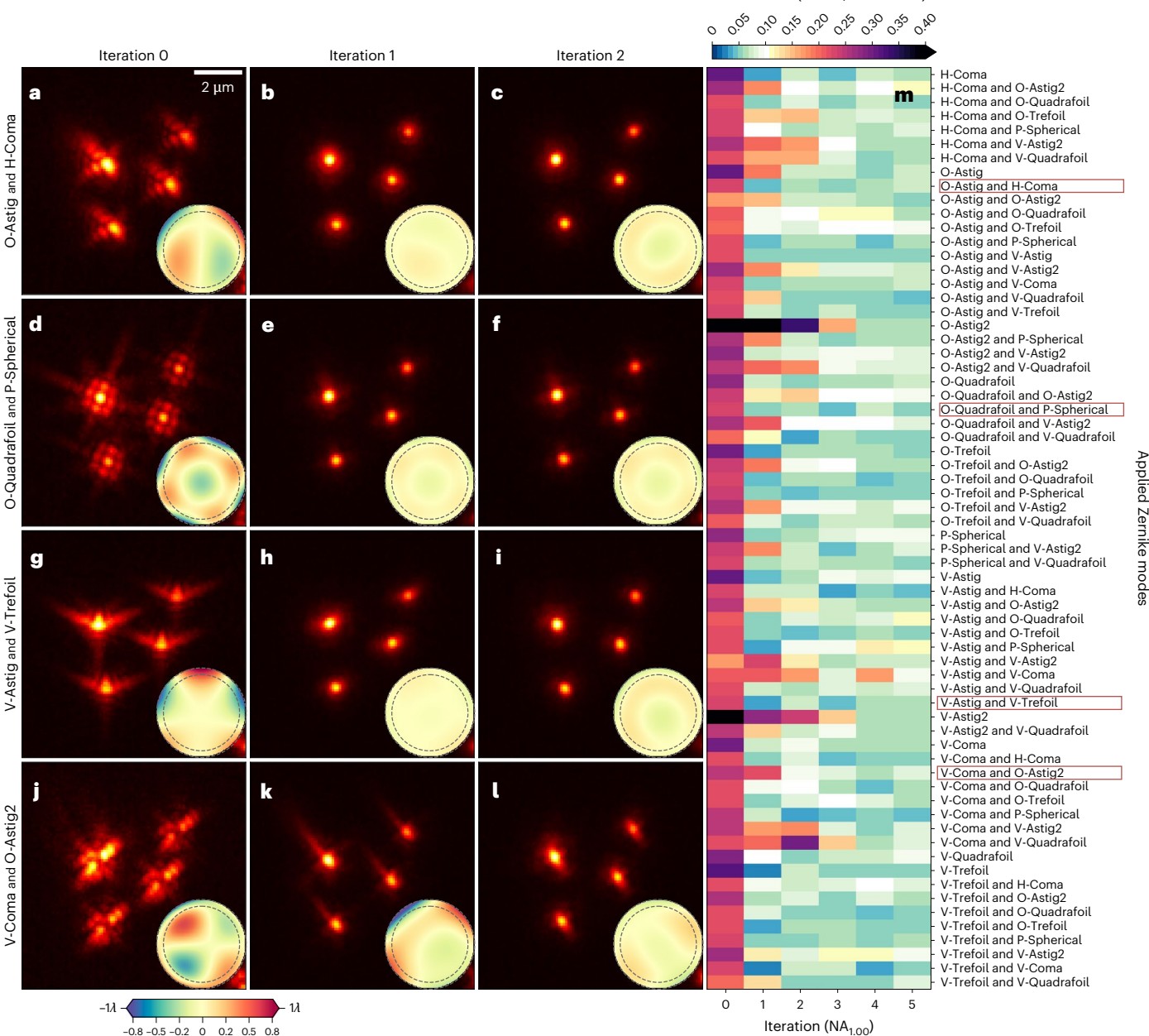

**Fig. 3 | Experimental correction of beads with initial artificial aberrations. a–l**, Four examples where the initial aberration was applied artificially by the DM. **a,b,c**, O-Astig and H-Coma ($Z_{n=2}^{m=-2} + Z_{n=3}^{m=1}$); **d,e,f**, O-Quadrafoil and P-Spherical ($Z_{n=4}^{m=-4} + Z_{n=4}^{m=0}$); **g,h,i**, V-Astig and V-Trefoil ($Z_{n=2}^{m=2} + Z_{n=3}^{m=-3}$); **j,k,l**, V-Coma and O-Astig2 ($Z_{n=3}^{m=-1} + Z_{n=4}^{m=-2}$). Iteration 0 shows XY maximum intensity projection (MIP) of four beads with initial aberration imaged using LLS, providing initial conditions for AOVIFT predictions (**a,d,g,j**). Iteration 1 shows the resulting field of beads after applying AOVIFT prediction to the DM (**b,e,h,k**). Iteration 2 shows the results after applying the AOVIFT prediction measured from Iteration 1 (**c,f,i,l**). Insets: the AOVIFT predicted wavefront over the NA = 1.0 pupil with a dashed line at NA = 0.85. **m**, Heatmap of the residual aberrations (measured by PhaseRetrieval on isolated bead) after applying AOVIFT predictions, starting with a single Zernike mode up to Mode 14 ($Z_{n=4}^{m=4}$) across up to five iterations.

corresponding to a Strehl ratio of 0.8 under the Rayleigh quarter-wave criterion[10,11]. In silico evaluation using the $10^4$ single punctum test images shows that AOVIFT recovers diffraction-limited performance in a single iteration in nearly all trials where the initial aberration is <0.30λ RMS and the integrated signal is >5 × $10^4$ photons (Fig. 2e). The corrective range increases to 0.40, 0.50, 0.55 and 0.6λ RMS for two to five iterations, respectively, although ~5 × $10^4$ photons remains the floor of required signal (Fig. 2f and Supplementary Fig. 10). This is comparable to the signal needed for SH wavefront sensing[2,3] and at least three times lower than that needed for PhaseRetrieval. In comparison, PhaseNet[4] and PhaseRetrieval[12] extend the diffraction-limited range only slightly (initial aberration <0.15λ RMS) after a single iteration on

the same test data (Supplementary Fig. 11a,b) and, in contrast to AOVIFT, do not appreciably increase this range after several iterations (Supplementary Fig. 12a–f). PhaseRetrieval does advantageously reduce residuals after a single iteration over a much broader range of initial aberration than AOVIFT, and this trend continues with further iteration, albeit never back to the diffraction limit (Supplementary Fig. 12a–c). However, this advantage is lost if the fiducial bead is not centered in the field of view (FOV), and the predictive power of PhaseNet is lost completely under the same circumstances (Supplementary Fig. 11d,e) because the widefield 3D image of the bead is then clipped. Furthermore, PhaseRetrieval and PhaseNet assume a priori the existence of only a single bead. AOVIFT is trained on one of five puncta falling anywhere

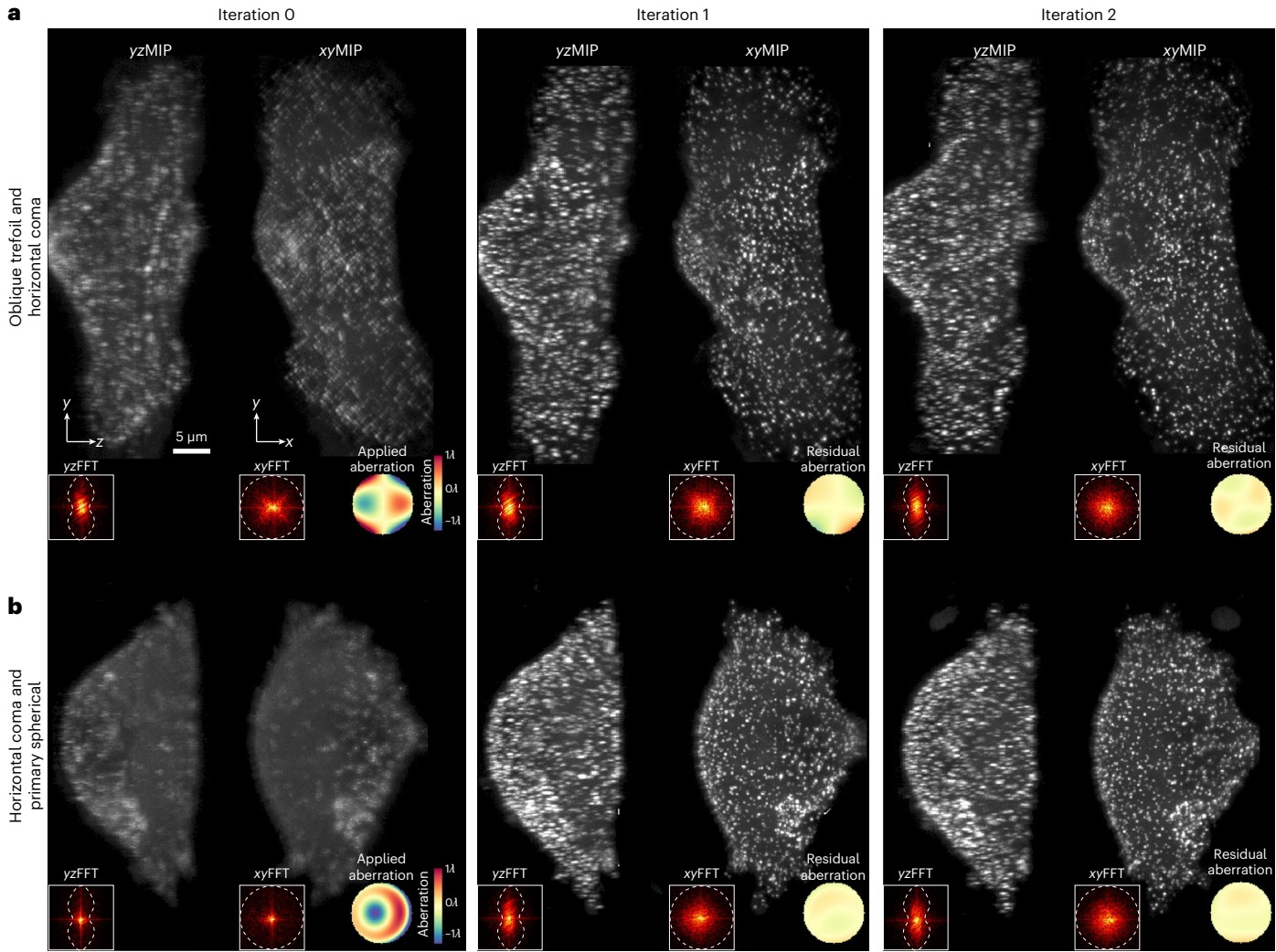

**Fig. 4 | Correction of aberrations in live SUM159-AP2 cells expressing $\sigma$2-eGFP.** **a**, 3D volume SUM159-AP2 cells represented as *xy*MIPs and *yz*MIPs covering a 15.7 × 55.6 × 25.6 μm³ FOV after applying a 2.9$\lambda$ P–V aberration to the DM. This aberration combines horizontal coma $Z_3^1$ and oblique trefoil $Z_3^3$. **b**, *xy*MIPs and *yz*MIPs of a similar FOV with 3.1$\lambda$ P–V aberration composed of horizontal coma ($Z_3^1$) and primary spherical ($Z_4^0$). In both cases, near diffraction-limited performance was recovered after two iterations. Insets: FFTs and corresponding wavefronts for each iteration.

within the FOV but, thanks to the normalization step to eliminate phase fringes from several puncta (Supplementary Fig. 24dd), produces inferences comparably accurate to a single punctum for up to 150 puncta, provided their mean nearest neighbor distance is >400 nm (Fig. 2g,h and Supplementary Fig. 13). Indeed, AOViFT relies on the combined signal of several native but dim subdiffractive biological assemblies such as CCPs to achieve accurate inferences.

**Experimental characterization on fiducial beads**

We performed all experiments using the AO-LLSM microscope schematized in Supplementary Fig. 14. For initial characterization of the ability of AOViFT to correct a wide range of possible aberrations, we performed 66 separate experiments wherein we:

(1) introduced aberration by applying to the deformable mirror (DM) one of the 66 possible combinations of one or two Zernike modes (from the first 15, excluding piston, tip/tilt and defocus), with each mode set to 0.2$\lambda$ RMS amplitude

(2) used AO-LLSM with the MBSq-35 LLS excitation profile of Supplementary Table 6 to image a field of 100 nm diameter fluorescent beads with this aberration;

(3) used AOViFT to predict the aberration

(4) applied the corrective pattern to the DM

(5) repeated (1)–(5) for five iterations

In 45 cases, we recovered diffraction-limited performance in two iterations (Fig. 3) and, in five iterations for 11 more cases (Supplementary Fig. 15). In the remaining ten cases, aberrations were reduced by at least 50% after five iterations.

**Correction of aberrations on live cultured cells**

We next tested the ability of AOViFT to correct aberrations during live cell imaging under biologically relevant conditions of limited signal, dense puncta and specimen motion. To this end, we applied aberrations to the DM and imaged cultured SUM159 human breast-cancer-derived cells gene edited to produce endogenous levels of the clathrin adapter protein AP2 tagged with eGFP. This yielded numerous membrane-bound CCPs at various stages of maturation that were suitable for aberration measurement. In one example (Fig. 4a), we applied a 2.9$\lambda$ peak-to-valley (P–V) aberration to the DM consisting of a mix of horizontal coma and oblique trefoil ($Z_3^1$ and $Z_3^3$), and recovered near diffraction-limited performance after two iterations (Supplementary Table 4). Peak signal at the CCPs increased twofold to three-fold postcorrection, and the spatial frequency content as seen in

orthoslices through the 3D fast Fourier transform (FFT) (insets at bottom) increased in every iteration. In another case (Fig. 4b), we reduced a 3.1$\lambda$ P–V aberration composed of a combination of horizontal coma and primary spherical ($Z_3^1$ and $Z_4^0$) to 0.069$\lambda$ RMS after two iterations, increasing CCP signal by threefold to fourfold (Supplementary Table 4). Four more examples of correction on cells and fiducial beads after applying single modes of 1$\lambda$ P–V aberration are given in Supplementary Fig. 16 and five more examples of two-mode correction are shown in Supplementary Fig. 17.

### In vivo correction of native aberration within a zebrafish embryo

As a transparent vertebrate, zebrafish are a popular model organism for imaging studies. However, the spatially heterogenous refractive index within multicellular organisms and the discontinuity of refractive index at their surfaces with respect to the imaging medium result in aberrations that vary throughout their interiors (Fig. 5a). We corrected a ~ 2$\lambda$ P–V aberration in one such region (Fig. 5b, top) with AOVIFT (Fig. 5b, bottom) near the notochord of a transgenic zebrafish embryo 72 h postfertilization expressing AP2-mNeonGreen in CCPs at the membranes of all cells (*ap2s1:ap2s1-mNeonGreenb$^{k800}$*; Methods) and recovered spatial frequencies across the corrected volume (FFTs at right) comparable to SH correction over the same region (Fig. 5b, middle).

In a second embryo expressing AP2-mNeonGreen in CCPs and mChilada-Cox8a in mitochondria (Fig. 5c), we used the mNeonGreen signal to correct a -1.5$\lambda$ P–V aberration (top row) in one region, which provided an aberration-corrected view of both CCPs and mitochondria (second row). Deconvolution of the aberrated images using an assumed ideal point spread function (PSF) amplified only high frequency artifacts (third row), but provided a more accurate representation of sample structure (bottom row) for the aberration-corrected ones by compensating for known attenuation of high spatial frequencies in the ideal optical transfer function (OTF).

### Correction of spatially varying aberrations in vivo

With GS-illuminated SH sensors, aberration measurement is not accurate unless it is confined to a single isoplanatic region. However, these are often much smaller than the volume of interest, and their boundaries are not generally known a priori. Consequently, microscopists are often forced to map aberrations by serial SH measurement over many small, tiled subregions whose dimensions are a matter of educated guesswork. On the other hand, with AOVIFT we generated a complete map of 204 aberrations (Fig. 6a) at 6.3-µm intervals over $37 \times 211 \times 12.8$ µm³ in a live zebrafish embryo 48 hpf (Fig. 6b,d) in -1.5 min on a single node of four A100 graphics processing units (GPUs). Unfortunately, it is not possible to apply a corrective pattern to a single pupil conjugate DM and thereby correct this spatially varying aberration across the entire FOV. One option would be to apply each aberration in turn and image the tiles one by one, or together in groups of similar aberration. Although slow, this would recover the full information of which the microscope is capable. However, a much faster and simpler alternative is to deconvolve each raw image tile with its own unique aberrated PSF (Fig. 6c,e). This does not recover full diffraction-limited performance, but it does suppress aberration-induced artifacts and provides a more faithful representation of the underlying sample structure (Fig. 6f–h).

## Discussion

AOVIFT provides accurate mapping of spatially varying sample-induced aberrations in specimens having subdiffractive puncta. Although AOVIFT can be slower than using SH for a single region of interest, it gains a substantial net speed advantage when mapping several regions of interest across a large FOV due to its parallelizable inference framework (Supplementary Table S5). Moreover, its throughput can be further accelerated by distributed GPU processing across several nodes and by compiling the model with TensorRT (https://docs.nvidia.com/deeplearning/tensorrt/pdf/TensorRT-Developer-Guide.pdf) for optimized inference. Unlike AOVIFT, SH measurement with a TPE GS has the key advantage of being agnostic to the fluorescence distribution within each isoplanatic region, but requires additional hardware (TPE laser, galvos, SH sensor) and the TPE power level must be carefully monitored to minimize photodamage. In addition, since the isoplanatic regions are not known a priori, the initial measurement grid for SH sensing must be very dense to accurately map aberrations and their rate of change across the FOV, requiring additional time at an additional cost to the photon budget. Conversely, AOVIFT determines the aberration map from a single large 3D image volume, and can therefore iteratively adjust tile sizes or positions in silico as needed until the map converges to an accurate solution.

Although trained for a specific LLS type (Supplementary Table 6), AOVIFT retained predictive capability when tested in silico with other light sheets as well (Supplementary Fig. 20). Although training specifically for such light sheets might increase the predictive range even further, a more fruitful path might be to augment the synthetic training data with light sheets axially offset from the detection focal plane to replace the closed-loop hardware-based mitigation of such offsets needed now[3]. Future models might leverage ubiquitous subcellular markers, such as plasma membranes or organelles, rather than genetically expressed diffraction-limited puncta, provided these markers contain sufficient high spatial frequency content for accurate inference of aberrations. Finally, to enhance generalizability of AOVIFT and reduce overfitting to narrowly defined imaging scenarios, future models should incorporate a more diverse range of light sheets, specimen types and labeling strategies.

Development of AOVIFT highlighted the challenges of constructing a 3D transformer-based architecture for AO correction. Each iteration of model design, training and testing required specialized simulated data pipelines, large GPU resources and extensive hyperparameter tuning—leading to lengthy model development cycles. A key bottleneck is the absence of universally applicable, large pretrained models for volumetric imaging data—a limitation that extends beyond AO applications.

Unlike the natural image domain, where ViT benefited from extensive training on standardized two-dimensional (2D) datasets, a comparable 'foundation model' for 3D microscopy is pending the collection of similar datasets. This gap severely limits how far and how quickly new methods like AOVIFT can be generalized. Although our work highlights the feasibility of building a solution for a given task (for example, AO corrections under specific imaging conditions), adapting to new scenarios such as new sample types, microscope geometries or aberration ranges typically requires substantial retraining and additional data curation.

These limitations highlight the need for pretrained foundation models in volumetric microscopy. We consider AOVIFT—a 3D vision

---

**Fig. 5 | In vivo, in situ correction of native aberrations in zebrafish embryos. a**, *xy*MIP of a 72-hpf gene-edited zebrafish embryo expressing endogenous AP2-mNeonGreen, exhibiting native and spatially varying aberrations near the notochord. **b**, Enlarged view of the dashed blue box in **a**. The *xy*MIPs and *yz*MIPs, along with the corresponding FFTs of a 12.5 × 12.5 × 12.8 µm³ FOV, show -2$\lambda$ P–V of sample-induced aberration without AO (top row), corrected by SH (middle row) and corrected by AOVIFT (bottom row). The contrast for each volume was scaled to its 1st and 99.99th percentile intensity values. **c**, *xy*MIPs and *yz*MIPs of a different gene-edited zebrafish embryo expressing exogenous AP2-mNeonGreen and injected mRNA for mChilada-Cox8a (to visualize mitochondria). The AP2 signal was used to infer the underlying aberration, and the same correction was applied to both channels. Top: -1.5$\lambda$ P–V aberration; middle: AOVIFT correction after two iterations. The third and fourth rows present the results of OMW deconvolution without and with AOVIFT corrected volumes, respectively.

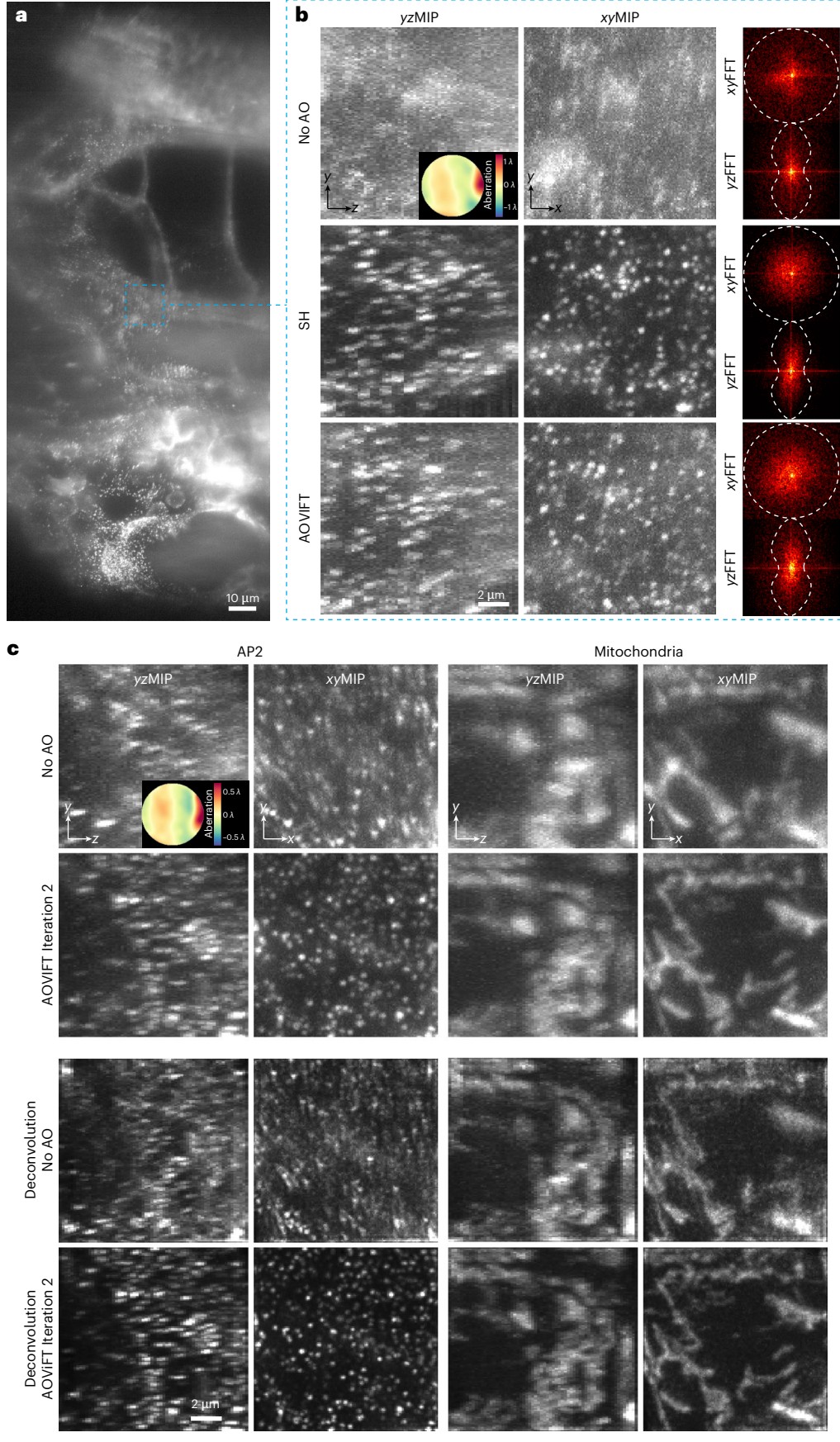

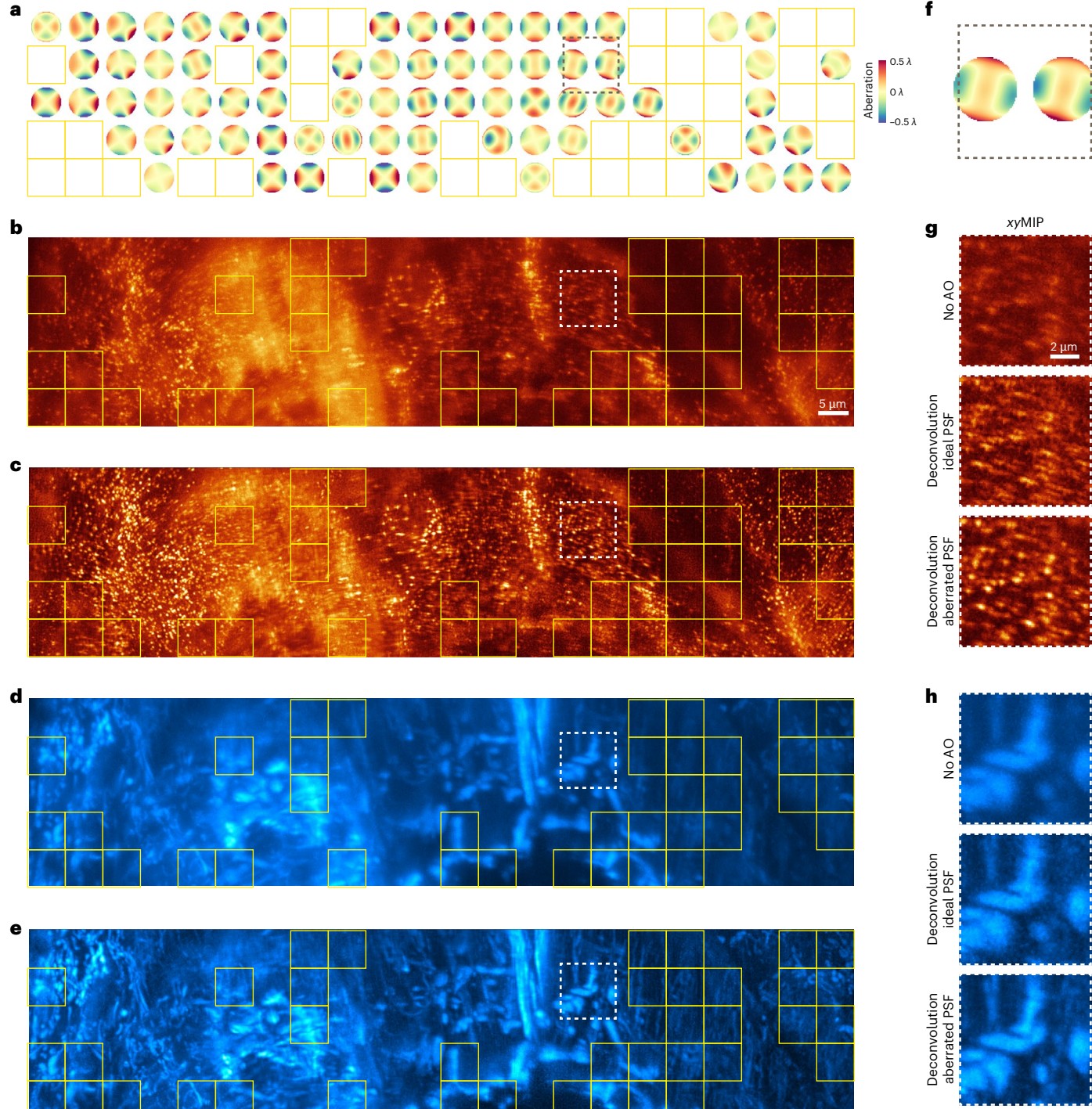

**Fig. 6 | Correcting aberrations postacquisition using spatially varying deconvolution in a zebrafish embryo. a**, Isoplanatic patch map determined by AOViFT for 204 tiles (6.3 × 6.3 × 12.8 μm³ each), spanning 37 × 211 × 12.8 μm³ FOV in a live, gene-edited zebrafish embryo expressing endogenous AP2-mNeonGreen. The yellow box marks areas with insufficient spatial features to accurately determine aberrations; an ideal PSF was used for OMW deconvolution in these regions. **b**, xyMIP of the AP2 signal without AO. **c**, xyMIP of each tile after deconvolution with spatially varying PSFs predicted by AOViFT. **d,e**, Raw (**d**) and deconvolved (**e**) xyMIPs of the mitochondria channel for the same region. **f**, Enlarged view of the wavefronts within the black dashed box in **a**. **g,h**, Zoomed-in views of AP2 (**g**) and mitochondria (**h**) structures from **b**–**e**, comparing No AO to OMW deconvolution using either an ideal PSF or spatially varying tile-specific aberrated PSFs predicted by AOViFT.

transformer model—as a stepping stone towards the more ambitious goal of creating a 4D model pretrained on massive volumetric microscopy datasets. Such a model could be fine-tuned for tasks across spatial (from molecules to organisms) and temporal (from stochastic molecular kinetics to embryonic development) scales[13]. Realizing this vision would require petabytes of high-quality curated 4D datasets and significant computational resources. However, successful implementation would dramatically shorten development timelines, improve generalization and reduce the overhead of custom training for varied experimental setups or microscope configurations.

## Online content

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

## Methods

### AO-LLS microscope

Imaging was performed using an AO-LLS microscope similar to one described previously[3] (Supplementary Fig. 14 and Supplementary Table 7). Briefly, 488-nm and 560nm lasers (500 mW 2RU-VFL-P-500-488-B1R and 1,000 mW 2RU-VFL-P-1000-560-B1R, MPB Communications Inc.) were modulated using an acousto-optical tunable filter (Quanta-Tech, AA OptoElectronic, AOTFnC-400.650-CPCh-TN) and shaped into a stripe by a Powell lens (Laserline Optics Canada, LOCP-8.9R20-2.0) and a pair of 50- and 250-mm cylindrical lenses (25-mm diameter; Thorlabs, ACY254-050 and LJ1267RM-A). The stripe illuminated a reflective, phase-only, gray-scale spatial light modulator (Meadowlark Optics, AVR Optics, P1920-0635-HDMI; 1,920 × 1,152 pixels) located at a sample conjugate plane. An eight-bit phase pattern written to the spatial light modulator generated the desired light-sheet pattern in the sample, and an annular mask (Thorlabs Imaging) at a pupil conjugate plane blocked unwanted diffraction orders before the light passed through the excitation objective (Thorlabs, TL20X-MPL). A pair of pupil conjugate galvanometer mirrors (Cambridge Technology, Novanta Photonics, 6SD11226 and 6SD11587) scanned the light sheet at the sample plane. The sample was positioned at the common foci of the excitation and detection objectives by a three-axis XYZ stage (Smaract; MLS-3252-S, SLS-5252-S and SLS-5252-S). Fluorescence emission from the sample was collected by a detection objective (Zeiss, ×20 1.0 numerical aperture (NA), 421452-9800-000), reflected off a pupil conjugate DM (ALPAO, DM69) that applied aberration corrections, and then recorded on two sample conjugate cameras (Hamamatsu ORCA Fusion).

SH measurements (Supplementary Fig. 18) were performed on the same microscope by localizing the intensity maxima (on a Hamamatsu ORCA Fusion) formed by the emitted light after passage through a pupil conjugate lenslet array (Edmund Optics, 64-479). The positional shifts of these maxima relative to those seen with no specimen present encode the pupil wavefront phase[2], which can then be reconstructed.

### Integration with microscope

AOViFT inference is performed routinely on the microscope acquisition PC (Intel Xeon, W5-3425, Windows11, 512 GB RAM, NVIDIA A6000 with 48 GB VRAM). Inferences are made in an Ubuntu Docker container based on the TensorFlow NGC Container (24.02-tf2-py3) running in parallel with the microscope control software. Data communication between AOViFT and the microscope control software is handled through the computer's file system. Image files and command-line parameters are passed to the model, and an output text file reports the resultant DM actuator values (Supplementary Fig. 19). When a volume is large enough to require tiling and dozens of volumes need to be processed, model inferences are parallelized and run using a SLURM compute cluster consisting of four nodes, each node containing four NVIDIA A100 80GB.

### Fluorescent beads and cells expressing fluorescent endocytic adapter AP2

The 25-mm coverslips (Thorlabs, CG15XH) used for imaging beads, cells and zebrafish embryos were first cleaned by sonication in 70% ethanol followed by Milli-Q water, each for at least 30 min. They were then stored in Milli-Q water until use. Gene-edited SUM159-AP2-eGFP cells[14] were grown in Dulbecco's modified Eagle's medium (DMEM)/F12 with GlutaMAX (Gibco, 10565018) supplemented with 5% fetal bovine serum (FBS; Avantor Seradigm, 89510-186), 10 mM HEPES (Gibco 15630080), 1 µg ml$^{-1}$ hydrocortisone (Sigma, H0888), 5 µg ml$^{-1}$ insulin (Sigma, I9278). Fluorescent beads (0.2-µm diameter, Invitrogen FluoSpheres Carboxylate-Modified Microspheres, 505/515 nm, cat. no. F8811 or 0.2-µm diameter Tetraspeck, Thermo Fisher Scientific Invitrogen, T7280) alone or with cells at 30–50% confluency were deposited onto plasma-treated and poly-D-lysine (Sigma-Aldrich, P0899)-treated 25-mm coverslips. Cells were cultured under standard conditions (37 °C, 5% CO$_2$, 100% humidity) with twice weekly passaging. The SUM159-AP2-eGFP cells were imaged in Leibovitz's L-15 medium without phenol red (Gibco,21083027) with 5% FBS (American Type Culture Collection, SCRR-30-2020), 100 µM Trolox (Tocris, 6002) and 100 µg ml$^{-1}$ Primocin (InvivoGen, ant-pm-1) at 37 °C. Aberrations of approximately 1λ P–V were induced using a DM in ten configurations of Zernike modes ($Z_2^2$, $Z_3^{-3}$, $Z_3^{-1}$, $Z_4^0$ and their pairwise combinations). Widefield PSFs were collected from 0.2-µm fluorescent beads to confirm the aberrations applied and residual aberrations after correction (Supplementary Table 7).

### Zebrafish embryos expressing fluorescent AP2 and mitochondria

Genome-edited *ap2s1*-expressing zebrafish (genome editing of *ap2s1*, *ap2s1:ap2s1-mNeonGreen*[bk800]; Supplementary Note D) were injected with cox8-mChilada mRNA for two color experiments. The N-terminal 34 amino acids of Cox8a were cloned into a pMTB backbone with a linker and mChilada coding sequence on the C terminus (unpublished, gift from N. Shaner). The plasmid was linearized, and mRNA was synthesized using a SP6 mMessage mMachine transcription kit (Thermo Fisher). RNA was purified using an RNeasy kit (Qiagen) and embryos were injected with 2 nl of 10 ng µl$^{-1}$ Cox8a-mChilada, 100 mM KCl, 0.1% phenol red, 0.1 mM EDTA and 1 mM Tris, pH 7.5. Zebrafish embryos were first nanoinjected with 3 nl of a solution containing 0.86 ng µl$^{-1}$ α-bungarotoxin protein, 1.43 × PBS and 0.14% phenol red. The injected embryos were mounted for imaging using a custom, volcano-shaped agarose mount. Each mount was constructed by solidifying a few drops of 1.2% (w/w) high-melting agarose (Invitrogen UltraPure Agarose, 16500–100, in 1× Danieau buffer) between a 25-mm glass coverslip and a 3D-printed mold (Formlabs Form 3+, printed in clear v.4 resin). This created ridges that formed a narrow groove. A hair-loop was used to orient the embryo within the agarose groove, positioning the left lateral side upward. Subsequently, 10–20 µl of 0.5% (w/w) low-melt agarose (Invitrogen UltraPure LMP Agarose, 16520–100, in 1× Danieau buffer) preheated to 40 °C, containing 0.2 µm Tetraspeck microspheres, was added on top of the embryo. This layer solidified around the embryo to secure it while providing fiducial beads for sample finding. Once the low-melt agarose solidified, the volcano-shaped mount was held by a custom sample holder for imaging. The embryo was oriented so that its anterior–posterior axis lay parallel to the sample *x* axis, with the anterior end facing the excitation objective and the posterior end facing the detection objective. The microscope objectives and the sample was immersed in a bath of ~50 ml bath Danieau buffer and were fully submerged, ensuring the embryo remained in buffered medium. Measurements for AOViFT and SH were done serially on the same FOV to compare the aberration corrections of both methods (Supplementary Table 7).

### Spatially varying deconvolution

To compensate for sample-induced aberrations postacquisition, we performed a tile-based spatially varying deconvolution on each 3D volume. Each volume was first subdivided into several 3D tiles approximating isoplanatic patches. A AOViFT predicted PSF (for compensation) or an ideal PSF (for no compensation) was assigned to each tile, and aberrations were corrected using OTF masked Wiener (OMW) deconvolution[15]. To minimize boundary artifacts during deconvolution, the tile size was extended by half the PSF width at each boundary (32 pixels); after deconvolution, these overlaps were removed and the deconvolved core regions were stitched together to form the final corrected volume. All computations were done in MATLAB v.2024a (Mathworks).

### Synthetic training/testing datasets

To train a model for predicting optical aberrations from images of subdiffractive objects in biological samples, we generated synthetic

datasets encompassing a range of relevant variables (for example, aberration modes and amplitudes, number and density of puncta, SNR). This synthetic dataset generation procedure is as follows.

For a single subdiffractive punctum, the electric field in the rear pupil of the detection objective is given by:

$$E(k_x, k_y) = A(k_x, k_y) \exp(i\phi(k_x, k_y)) \tag{1}$$

where $A(k_x, k_y)$ is the pupil amplitude with coordinates $k_x$, $k_y$, and $\phi(k_x, k_y)$ is the pupil phase. Under aberration-free conditions, $\phi(k_x, k_y)$ is constant. We can empirically determine $A(k_x, k_y)$ by acquiring a widefield image of an isolated subdiffractive object (100-nm fluorescent bead), performing phase retrieval[12,16] and applying the opposite of the retrieved phase using a pupil conjugate DM so that $\phi(k_x, k_y)$ becomes a constant.

The electric field for the image of a single aberrated punctum is:

$$E_{abb}(k_x, k_y) = A(k_x, k_y) \exp(i\phi_{abb}(k_x, k_y)) \tag{2}$$

where the $\phi_{abb}(k_x, k_y)$ is described as a weighted sum of Zernike modes of unique amplitudes:

$$\phi_{abb}(k_x, k_y) = \sum_{m,n} \alpha_n^m Z_n^m(k_x, k_y) \tag{3}$$

Empirically, zebrafish induced aberrations for the microscopes used here are well described by combinations of 11 of the first 15 Zernike modes[17] (Supplementary Fig. 1), for which $n \leq 4$, excluding piston ($Z_0^0$), tip ($Z_1^{-1}$), tilt ($Z_1^1$) and defocus ($Z_2^0$) (as these represent phase offsets or sample translation). The distributions and amplitudes of the remainder are used to build the training set as discussed below.

The aberrated 3D detection PSF of a subdiffractive punctum is approximated by:

$$PSF_{abb}^{det}(x,y,z) = \left| \iint_{pupil} E_{abb}(k_x, k_y) \exp[i(k_x x + k_y y + k_z z)] dk_x dk_y \right|^2 \tag{4}$$

where $k_z = \sqrt{\left(\frac{2\pi\eta}{\lambda}\right)^2 - k_x^2 - k_y^2}$, $\eta$ is the refractive index of the imaging medium and $\lambda$ is the free-space wavelength of the fluorescence emission.

For light sheet microscopy, the aberrated 3D overall PSF is:

$$PSF_{abb}^{overall}(x,y,z) = PSF^{exc}(z) \cdot PSF_{abb}^{det}(x,y,z) \tag{5}$$

where $PSF^{exc}(z)$ is given by the cross-section of the swept light sheet used for imaging. Examples of these PSFs are shown in Supplementary Fig. 20 I–V, with MBSq-35 in Supplementary Table 6 used for training and imaging (see ref. 18 for additional information on these light sheets).

Each synthetic training volume sample $V$ is $64 \times 64 \times 64$ voxels in size spanning $8 \times 8 \times 12.8\ \mu m^3$ (with $125 \times 125 \times 200\ nm^3$ voxels) and containing between $J = 1$ to $J = 5$ puncta chosen from a uniform distribution and located randomly at points $(x_j, y_j, z_j)$ within the volume. Each punctum is modeled as a Gaussian of full width at half maximum $w_j$ chosen randomly from the set [100, 200, 300, 400] nm, allowing for slightly larger than the diffraction-limit features. The image of each punctum is generated by its convolution with the aberrated PSF:

$$I_j^{bead}(x,y,z) = PSF_{abb}^{overall}(x,y,z) \otimes \exp\left[-4\ln(2)\frac{x^2 + y^2 + z^2}{w_j^2}\right] \tag{6}$$

The integrated photons $N_o$ per punctum were selected from a uniform distribution of 1 to 200,000 photons. The total intensity distribution is:

$$I_{photon}(x,y,z) = \Upsilon \cdot \sum_{j=1}^{J} I_j^{bead}(x - x_j, y - y_j, z - z_j) \tag{7}$$

where

$$\Upsilon = \frac{N_o}{\iiint_{-\infty}^{\infty} I_j^{bead}(x,y,z)\, dx\, dy\, dz} \tag{8}$$

As the signal from each aberrated punctum can exceed the boundary of $V$, total signal $S_V$ within $V$ is:

$$S_V = \iiint_V I(x,y,z)\, dx\, dy\, dz \leq JN_o \tag{9}$$

After accounting for partial signal contributions ($S_v$) the photons per voxel were converted to camera counts by applying the quantum efficiency QE, Poisson shot noise $\eta$ and camera read noise $\epsilon$ to arrive at the final synthetic training set example:

$$I_{camera}(x,y,z) = QE \cdot I_{photon}(x,y,z) + \eta[QE \cdot I_{photon}(x,y,z)] + \epsilon \tag{10}$$

**Zernike distributions.** To ensure diversity in the training set to cover potential aberrations, each training example was chosen from the amplitudes of the 11 included aberration modes shown in color in Supplementary Fig. 1 with equal probability from one of four different distributions:

(1) Single mode (Supplementary Fig. 21b) One mode is randomly chosen, with amplitude $\alpha$ chosen randomly from $0 \leq \alpha \leq 0.5\lambda$ RMS.
(2) Bimodal (Supplementary Fig. 21c) An initial target for the total amplitude $\alpha_t$ is chosen randomly from $0 \leq \alpha_t \leq 0.5\lambda$ RMS. A second partitioning factor $\epsilon$ is chosen randomly from $0 \leq \epsilon \leq 1$. The amplitudes of the two modes are then $\alpha_1 = \epsilon\alpha_t$ and $\alpha_2 = (1 - \epsilon)\alpha_t$.
(3) Powerlaw (Supplementary Fig. 21d) An initial target for the total amplitude $\alpha_t$ is chosen randomly from $0 \leq \alpha_t \leq 0.5\lambda$ RMS. The initial partitioning factors $\epsilon_n$ for the modes are chosen randomly from a Lomax (that is, Pareto II) distribution[19]:

$$\epsilon_n = \frac{\gamma}{(x_n + 1)^{\gamma+1}} \quad \text{where} \quad \gamma = 0.75 \tag{11}$$

where each $x_n$ is chosen randomly from $0 \leq x_n \leq 1$. They are then renormalized:

$$\epsilon_n' = \frac{\epsilon_n}{\sum_{n=1}^{11} \epsilon_n} \tag{12}$$

and the final amplitudes of the modes are $\alpha_n = \epsilon_n'\alpha_t$.

(4) Dirichlet (Supplementary Fig. 21e) An initial target for the total amplitude $\alpha_t$ is chosen randomly from $0 \leq \alpha_t \leq 0.5\lambda$ RMS. The initial partitioning factors $\epsilon_n$ for the modes are chosen randomly from $0 \leq \epsilon_n \leq 1$. They are then renormalized:

$$\epsilon_n' = \frac{\epsilon_n}{\sum_{n=1}^{11} \epsilon_n} \tag{13}$$

and the final amplitudes of the modes are $\alpha_n = \epsilon_n'\alpha_t$.

Together, the training examples from these four distributions create a diverse set of overall aberration amplitudes and number of significant modes in the training data, with all 11 modes contributing equally across the dataset (Supplementary Fig. 21a).

**Training dataset.** For the model training, a dataset of 2 million synthetic 3D volumes was created, with aberration magnitude uniform sampled from 0.0 to $0.5\lambda$ RMS (at wavelength $\lambda = 510$ nm), uniform distribution of the number of objects between 1 and 5, and photons ranging between 1 and 200,000 integrated photons per object.

**Test dataset.** To evaluate our models, we created a test dataset with 100,000 3D volumes. The parameter distribution was the same as training, but extended the aberration magnitude up to $1.0\,\lambda$ RMS, and up to 500,000 integrated photons. To test the operational limit of our models, this test dataset included up to 150 objects in any given volume.

## Fourier embedding

Most ML vision models operate on real-space representations of the data, which lack clearly defined limits on image size or feature descriptors of their content. Instead, we used Fourier domain embeddings (Supplementary Fig. 24). These are bound by the microscope's OTF. Aberrations within an isoplanatic patch globally effect all photons within that patch, producing a unique, learnable 'fingerprint' pattern in the FFT amplitude and phase (Supplementary Note A.1 and Supplementary Figs. 5–7).

**Preprocessing.** To create Fourier embeddings (Fig. 1b) for our model, we preprocess the input 3D image stack $W$ of CCPs within an isoplanatic region to suppress noise and edge artifacts (Fig. 1a),

$$V = \Upsilon(W). \tag{14}$$

The preprocessing module ($\Upsilon$) begins with a set of filters to extract sharp-edged objects that reveal the aberration signatures: a Gaussian high-pass filter to remove inhomogeneous background and a low-pass filter through a Fourier frequency filter, with cutoff set at the detection NA limit ($\sigma = 3$ voxels). A Tukey window (Tukey cosine fraction = 0.5, in $\hat{x}\hat{y}$ only) is applied to remove FFT edge artifacts from the volume borders. No windowing is applied along the axial direction, $\hat{z}$, because embeddings are constructed near $k_z = 0$ where aberration information is maximized.

**Embedding.** Once preprocessed, a ratio of the resultant 3D FFT amplitude, to the 3D FFT amplitude of the ideal PSF (undergoing identical preprocessing steps) is used to generate the amplitude embedding, $\alpha(k_z)$ at each $k_z$ plane:

$$V_{\text{ideal}} = \Upsilon(\text{PSF}_{\text{ideal}}) \tag{15}$$

$$\alpha = \frac{|\mathcal{F}(V)|}{|\mathcal{F}(V_{\text{ideal}})|} \tag{16}$$

where $\mathcal{F}$ denotes the 3D Fourier transform. The most useful information content is located at $k_z = 0$, the principal plane located at the midpoint of the $\hat{k_z}$-axis. Three 2D planes from $\alpha_1$, $\alpha_2$ and $\alpha_3$ along $\hat{k_z}$-axis are necessary to extract axial information for inputs to the model as follows:

$$\alpha_1 = \alpha_{k_z=0} \tag{17}$$

$$\alpha_2 = \frac{1}{5} \sum_{i=0}^{4} \alpha_{k_z=i} \tag{18}$$

$$\alpha_3 = \frac{1}{5} \sum_{i=5}^{9} \alpha_{k_z=i} \tag{19}$$

where $\alpha_1$ is the principal plane along the $k_x$-axis and $k_y$-axis, $\alpha_2$ is the mean of five consecutive 2D planes starting from the principal plane and $\alpha_3$ is the mean of five consecutive 2D planes starting from the $k_z = 5$ plane (Supplementary Figs. 7 and 24a,c).

For the phase embedding, $\varphi$, we first remove interference from several puncta in the FOV that may obscure the aberration signature in the phase image. The interference patterns are removed using: peak local maxima (PLM; https://scikit-image.org/docs/stable/auto_examples/segmentation/plot_peak_local_max.html) for peak detection in

real space using normalized cross-correlation (NCC; https://scikit-image.org/docs/stable/auto_examples/registration/plot_masked_register_translation.html) with a kernel cropped from the highest peak in $V$. The neighboring voxels around the detected puncta peaks are masked off, creating a volume, $\mathcal{S}$. The OTF with interference removed, $\tau'$, can now be obtained as well as a real space reconstructed volume, $V'$, through inverse FFT,

$$M = \text{PLM}(\text{NCC}(V)) \tag{20}$$

$$\mathcal{S} = V \times M \tag{21}$$

$$\tau = \frac{\mathcal{F}(V)}{\mathcal{F}(\mathcal{S})} \tag{22}$$

$$V' = \mathcal{F}^{-1}(\tau) \tag{23}$$

The phase $\varphi(k_z)$ at each $k_z$ plane is then given by the unwrapped phase of $\tau$ at that plane (Supplementary Fig. 24b,d). We calculate the three phase embeddings in the same manner as our amplitude embedding such that:

$$\varphi_1 = \varphi_{k_z=0} \tag{24}$$

$$\varphi_2 = \frac{1}{5} \sum_{i=0}^{4} \varphi_{k_z=i} \tag{25}$$

$$\varphi_3 = \frac{1}{5} \sum_{i=5}^{9} \varphi_{k_z=i} \tag{26}$$

Combining the six planes together, we define the input to the model as a Fourier embedding,

$$\mathcal{E} = \{\alpha_1, \alpha_2, \alpha_3, \varphi_1, \varphi_2, \varphi_3\} \tag{27}$$

A notable advantage of this approach is that, although the signal from each individual CCP is weak, those in the same isoplanatic region contain near-identical spatial frequency distributions that add together to yield Fourier embeddings of high SNR suitable for accurate inference of the underlying aberration (Supplementary Fig. 6).

## AO vision Fourier transformer

Below, we outline the key components of AOVɪFT, which uses a 3D multistage vision transformer architecture. This model efficiently captures Fourier domain features at several spatial scales, enabling robust aberration prediction.

**Multistage.** Recent advances in attention-based transformers have demonstrated scalability, generalizability and multi-modality for a range of computer vision applications[20–24].

Multiscale (or hierarchical) vision transformers, such as Swin[25] and MViT[26], are designed with specialized modules (for example, shifted-window partitioning[25] and hybrid window attention[27]) to excel at a variety of detection tasks for 2D natural images using supervised training on ImageNet[28]. Although these variants are more efficient than their ViT counterparts in terms of FLOPs and number of parameters, they often incorporate specialized modules as noted above. Hiera[29] showed that these designs can be streamlined without performance loss by leveraging large-scale self-supervised pretraining.

Current multiscale architectures use a feature pyramid network scheme[30]—downsampling the spatial resolution of the image for each stage while expanding the embedding size for deeper layers. Instead, in our work, we use $\Omega$ stages and do not downsample during any of the

stages, but rather select different patch sizes for each stage (Fig. 1). This allows the embedding dimension within each stage to be fixed to the number of voxels in the patch of that stage, rather than expanding with increasing depth as in some hierarchical models.

**Patch encoding.** The input to the model is the Fourier embedding, a 3D tensor $\varepsilon \in \mathbb{R}^{\ell \times d \times d}$, where $\ell = 6$ is the number of 2D planes each with a height and width of $d$. For each model stage, $i$, patchifying begins by dividing the input tensor $\varepsilon$ into nonoverlapping 2D tiles (each $p_i \times p_i$) that are each flattened into a one-dimensional patch for a total of $k_i$ patches in a plane. After patchifying, the input tensor is transformed into $x_p \in \mathbb{R}^{\ell \times k_i \times p_i^2}$ (Fig. 1b).

The initial ViT model uses a set of consecutive transformer layers with a fixed patch size for all transformers, where each transformer layer can capture local and global dependencies between patches through self-attention[20]. The computation needed for the self-attention layers scales quadratically with reference to the number of patches (that is, sequence length). Although using a smaller patch size could be useful to capture visual patterns at a finer resolution, using a large patch size is computationally cheaper.

Our baseline model uses a two-stage design with patch sizes of 32 and 16 pixels, respectively (Fig. 1c). Supplementary Note A shows an ablation study using several stages with patch sizes ranging between 8 and 32 pixels.

**Positional encoding.** Rather than adopting the Cartesian positional encoding of ViT[20], we use a polar coordinate system $(r, \theta)$ to encode the position of each patch. This choice is motivated by the radial symmetries of the Zernike polynomials and the efficiencies gained in NeRF[31], coordinate-based MLPs[32] and RoFormer[33]. For a given plane in $\varepsilon$ (Eq. 27), the radial positional encoding vector (RPE) is calculated for every patch,

$$\text{RPE}(r, \theta) = [r, \sin\theta, \cos\theta, \dots, \sin m\theta, \cos m\theta] \tag{28}$$

where $(r, \theta)$ are the polar coordinates for the center of each patch, and $m = 16$. All patches and their positional encoding are then mapped into a sequence of learnable linear projections $\zeta \in \mathbb{R}^{\ell \times k_i \times p_i^2}$ that we use as our input to the transformer layers in the model.

**Transformer building blocks.** Each stage has $n$ transformer layers, where each layer has $h$ multihead attention (MHA) layers that map the interdependencies between patches, followed by a multilayer perceptron block (MLP) that learns the relationship between pixels within a patch. The stage's embedding size, $\epsilon_i = p_i^2$, is set to match the number of voxels in a patch for that stage. The MLP block is four times wider than the embedding size (Supplementary Fig. 2c). Layer normalization (LN)[34] is applied before each step, and a skip/residual connection[35] is added after each step:

$$\zeta_1 = \text{LN}(\text{MHA}(\zeta)) + \zeta \tag{29}$$

$$\zeta_2 = \text{LN}(\text{MLP}(\zeta_1)) + \zeta_1 \tag{30}$$

In addition to the skip connections in each transformer layer, we also add a skip connection between the input and output of each stage. We use a dropout rate of 0.1 for each dense layer[36] and stochastic depth rate of 0.1 (ref. 37). The patches from the final stage are pooled using a global average along the last dimension and passed to a fully connected layer to output $z$ Zernike coefficients.

**Attention modules.** We use self-attention[38] as our default attention module for all transformer layers in our model. Complementary to our approach, recent studies have looked into alternative attention methods to reduce the quadratic scaling of self-attention[23,39,40].

Our architecture is compatible with these attention mechanisms, which would further improve our model's efficiency.

## In silico evaluations

Supplementary Note A shows an ablation study of our synthetic data simulator (Supplementary Note A.2), our multistage design (Supplementary Note A.3), our training dataset size (Supplementary Note A.4 and Supplementary Fig. 2) and details of our training hyperparameters (Supplementary Note A.5, Supplementary Table 2 and Supplementary Table 8). We also introduce a new way of measuring prediction confidence of our model using digital rotations in Supplementary Note A.6 (Supplementary Fig. 23).

We present a detailed cost analysis benchmark comparing our architecture with other widely used models such as ConvNeXt[41] and ViT[20] in Supplementary Note B. To further diagnose our model's performance, we carried out a series of experiments to understand our model's sensitivity to SNR (Supplementary Note C.1 and Supplementary Figs. 24–25), generalizability to other light sheets (Supplementary Note C.2), number of objects in the FOV (Supplementary Note C.3) and object size (Supplementary Note C.4 and Supplementary Fig. 26).

## Ethics approval and consent to participate

All experiments with zebrafish were done in accordance with protocols approved by the University of California, Berkeley's Animal Care and Use Committee and following standard protocols (animal use protocol number AUP-2019-09-12560-1). All zebrafish used in this study were embryos younger than 72 h postfertilization. Sex determination was not a factor in our experiments. All husbandry and experiments with zebrafish were done in accordance with protocols approved by the University of California, Berkeley's Animal Care and Use Committee and following standard protocols (animal use protocol numbers AUP-2019-09-12560-1 (Upadhyayula laboratory), AUP-2020-10-13737-1 (Swinburne laboratory) and AUP-2021-05-14347-1 (Zebrafish Facility Core Protocol)).

## Reporting summary

Further information on research design is available in the Nature Portfolio Reporting Summary linked to this article.

## Data availability

Data for demos is available on our Github repository at https://github.com/cell-observatory/aovift. The full datasets for training and testing are too large to be hosted on public repositories, they can be shared upon reasonable request. Our synthetic data generator is also available at https://github.com/cell-observatory/beads_simulator to enable users to simulate their own datasets for training and evaluation.

## Code availability

Source code for training and evaluation (and all pretrained models) are available at https://github.com/cell-observatory/aovift. Docker image is available at https://github.com/cell-observatory/aovift/pkgs/container/aovift. Deconvolution was performed using PetaKit5D (https://github.com/abcucberkeley/PetaKit5D).

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

## Acknowledgements

We thank X. Ruan, M. Mueller, P. Zwart and H. York for helpful discussions and comments. SUM159 cells used in this study were a gift from the Kirchhausen laboratory. We thank N. Shaner for providing the mChilada fluorescent protein plasmid to I.A.S., which was used to generate reagents for this study. We gratefully acknowledge the support of this work by the Laboratory Directed Research and Development (LDRD) Program of Lawrence Berkeley National Laboratory under US Department of Energy contract no. DE-AC02-05CH11231. We thank J. White for managing our computing cluster. T.A., G.L., F.G., J.L.H. and S.U. are partially supported by the Philomathia Foundation (awarded to E.B. and S.U.). T.A. and G.L. are partially supported by the Chan Zuckerberg Initiative (awarded to S.U.). T.A. and S.U. are partially supported by Lawrence Berkeley National Laboratory's LDRD program 7647437 and 7721359 (awarded to S.U.). T.A., D.E.M. and E.B. are supported by HHMI (awarded to E.B.). C. Shirazinnejad. and D.G.D. are partially supported by NIH Grant R35GM118149 (awarded to D.G.D.). C. Simmons, I.S.A. and I.A.S. are partially supported by NIH Grant 1R01DC021710 (awarded to I.A.S.). F.G. is partially funded by the Feodor Lynen Research Fellowship, Humboldt Foundation. S.U. is funded by the Chan Zuckerberg Initiative Imaging Scientist program 2019-198142 and 2021-244163. E.B. is an HHMI Investigator. S.U. is a Chan Zuckerberg Biohub–San Francisco Investigator.

## Author contributions

T.A. designed models, developed training pipelines and evaluation benchmarks. D.E.M. designed the preprocessing algorithms and developed the microscope software for the imaging experiments. T.A. and D.E.M. designed Fourier embedding, and developed the synthetic data generator for training and validation. C. Shirazinnejad, C. Simmons, I.S.A., S.E.W., N.H., E. Hong, E. Huang, E.S.B., A.N.K., D.G.D. and I.A.S. generated the zebrafish reagents. A.N.K. prepared the cultured SUM159 cells. C. Shirazinnejad, J.L.H., K.A. and A.M.-J. prepared samples. G.L. and J.L.H. performed the imaging experiments with zebrafish. J.L.H. and K.A. performed the imaging experiments with cells. G.L., K.A. and F.G. performed the imaging experiments with beads. T.A., D.E.M. and S.U. performed analysis and prepared figures. T.A. wrote the paper with input from all co-authors. D.E.M., E.B. and S.U. edited the paper. T.A., E.B. and S.U. supervised the project.

## Competing interests

The authors declare no competing interests.

## Additional information

**Correspondence and requests for materials** should be addressed to Thayer Alshaabi, Srigokul Upadhyayula or Eric Betzig.

# Reporting Summary

## Statistics

For all statistical analyses, confirm that the following items are present in the figure legend, table legend, main text, or Methods section.

| n/a | Confirmed | |
|---|---|---|
| ☐ | ☒ | The exact sample size (*n*) for each experimental group/condition, given as a discrete number and unit of measurement |
| ☐ | ☒ | A statement on whether measurements were taken from distinct samples or whether the same sample was measured repeatedly |
| ☒ | ☐ | The statistical test(s) used AND whether they are one- or two-sided *Only common tests should be described solely by name; describe more complex techniques in the Methods section.* |
| ☒ | ☐ | A description of all covariates tested |
| ☒ | ☐ | A description of any assumptions or corrections, such as tests of normality and adjustment for multiple comparisons |
| ☒ | ☐ | A full description of the statistical parameters including central tendency (e.g. means) or other basic estimates (e.g. regression coefficient) AND variation (e.g. standard deviation) or associated estimates of uncertainty (e.g. confidence intervals) |
| ☒ | ☐ | For null hypothesis testing, the test statistic (e.g. *F*, *t*, *r*) with confidence intervals, effect sizes, degrees of freedom and *P* value noted *Give P values as exact values whenever suitable.* |
| ☒ | ☐ | For Bayesian analysis, information on the choice of priors and Markov chain Monte Carlo settings |
| ☒ | ☐ | For hierarchical and complex designs, identification of the appropriate level for tests and full reporting of outcomes |
| ☒ | ☐ | Estimates of effect sizes (e.g. Cohen's *d*, Pearson's *r*), indicating how they were calculated |

*Our web collection on statistics for biologists contains articles on many of the points above.*

## Software and code

Policy information about availability of computer code

| Data collection | LabVIEW software for microscope control, and code to generate the synthetic data used for training and evaluation are available at https://github.com/cell-observatory/aovift. |
|---|---|
| Data analysis | Code for training and evaluation (and all pretrained models) are available at https://github.com/cell-observatory/aovift. Deconvolution was performed using PetaKit5D https://github.com/abcucberkeley/LLSM5DTools Matlab (R2024a, MathWorks), Python (3.10), FIJI (1.53t). |

For manuscripts utilizing custom algorithms or software that are central to the research but not yet described in published literature, software must be made available to editors and reviewers. We strongly encourage code deposition in a community repository (e.g. GitHub). See the Nature Portfolio guidelines for submitting code & software for further information.

## Data

Policy information about availability of data

All manuscripts must include a data availability statement. This statement should provide the following information, where applicable:
- Accession codes, unique identifiers, or web links for publicly available datasets
- A description of any restrictions on data availability
- For clinical datasets or third party data, please ensure that the statement adheres to our policy

Data for demons and code for training and evaluation (and all pretrained models) are available at https://github.com/cell-observatory/aovift.

# Research involving human participants, their data, or biological material

Policy information about studies with human participants or human data. See also policy information about sex, gender (identity/presentation), and sexual orientation and race, ethnicity and racism.

| | |
|---|---|
| Reporting on sex and gender | N/A |
| Reporting on race, ethnicity, or other socially relevant groupings | N/A |
| Population characteristics | N/A |
| Recruitment | N/A |
| Ethics oversight | N/A |

Note that full information on the approval of the study protocol must also be provided in the manuscript.

# Field-specific reporting

Please select the one below that is the best fit for your research. If you are not sure, read the appropriate sections before making your selection.

☒ Life sciences ☐ Behavioural & social sciences ☐ Ecological, evolutionary & environmental sciences

For a reference copy of the document with all sections, see nature.com/documents/nr-reporting-summary-flat.pdf

# Life sciences study design

All studies must disclose on these points even when the disclosure is negative.

| | |
|---|---|
| Sample size | All sample sizes are described in the manuscript. |
| Data exclusions | No data were excluded in the manuscript. |
| Replication | All replications are described in the manuscript. |
| Randomization | N/A |
| Blinding | N/A |

# Reporting for specific materials, systems and methods

We require information from authors about some types of materials, experimental systems and methods used in many studies. Here, indicate whether each material, system or method listed is relevant to your study. If you are not sure if a list item applies to your research, read the appropriate section before selecting a response.

## Materials & experimental systems

| n/a | Involved in the study |
|---|---|
| ☒ | ☐ Antibodies |
| ☐ | ☒ Eukaryotic cell lines |
| ☒ | ☐ Palaeontology and archaeology |
| ☐ | ☒ Animals and other organisms |
| ☒ | ☐ Clinical data |
| ☒ | ☐ Dual use research of concern |
| ☒ | ☐ Plants |

## Methods

| n/a | Involved in the study |
|---|---|
| ☒ | ☐ ChIP-seq |
| ☒ | ☐ Flow cytometry |
| ☒ | ☐ MRI-based neuroimaging |

# Eukaryotic cell lines

Policy information about cell lines and Sex and Gender in Research

| | |
|---|---|
| Cell line source(s) | The AP2 Cells were shared by the Kirchhausen lab and published. (https://doi.org/10.1091/mbc.E16-03-0164) |

| Authentication | No further authentication was performed for this study. |
| Mycoplasma contamination | The cell lines were tested for mycoplasma contamination and the results were negative. |
| Commonly misidentified lines (See ICLAC register) | None of the cell lines used here belongs to commonly misidentified lines. |

# Animals and other research organisms

Policy information about studies involving animals; ARRIVE guidelines recommended for reporting animal research, and Sex and Gender in Research

| Laboratory animals | Genome edited Zebrafish with endogenous expression of ap2s1-mNeonGreen |
| Wild animals | No wild animals were used in this study. |
| Reporting on sex | sex was not considered in this study, the imaging was limited to developing zebrafish embryos. |
| Field-collected samples | N/A |
| Ethics oversight | All zebrafish experiments were conducted at UC Berkeley in accordance with the US National Institutes of Health Guide for the Care and Use of Laboratory Animals. Procedures and protocols were approved by the Institutional Animal Care and Use Committee of the UC Berkeley. |

Note that full information on the approval of the study protocol must also be provided in the manuscript.

# Plants

| Seed stocks | N/A |
| Novel plant genotypes | N/A |
| Authentication | N/A |

