## [Peer Review file · Nature Methods]

Fourier-Based 3D Multistage Transformer for Aberration Correction in Multicellular Specimens

Corresponding Author: Dr Thayer Alshaabi

Version 0:

Decision Letter:

29th Apr 2025

Dear Thayer,

Your Article, "Fourier-Based 3D Multistage Transformer for Aberration Correction in Multicellular Specimens", has now been seen by three reviewers. As you will see from their comments below, although the reviewers find your work of considerable potential interest, they have raised a number of concerns. We are interested in the possibility of publishing your paper in Nature Methods, but would like to consider your response to these concerns before we reach a final decision on publication.

We therefore invite you to revise your manuscript to address these concerns. In addition to addressing the technical concerns about performance and questions, we ask that you focus your revision on clarifying the sample space in which the methods works well, for the biologist reader.

Link Redacted

We hope to receive your revised paper within three months. If you cannot send it within this time, please let us know. In this event, we will still be happy to reconsider your paper at a later date so long as nothing similar has been accepted for publication at Nature Methods or published elsewhere.

OPEN SCIENCE REQUIREMENTS

REPORTING SUMMARY AND EDITORIAL POLICY CHECKLISTS

EXTENDED DATA FIGURES

DATA AVAILABILITY

All novel DNA and RNA sequencing data, protein sequences, genetic polymorphisms, linked genotype and phenotype data, gene expression data, macromolecular structures, and proteomics data must be deposited in a publicly accessible database, and accession codes and associated hyperlinks must be provided in the "Data Availability" section.

CODE AVAILABILITY

Please include a "Code Availability" subsection in the Online Methods which details how your custom code is made available. Only in rare cases (where code is not central to the main conclusions of the paper) is the statement "available upon request" allowed (and reasons should be specified).

MATERIALS AVAILABILITY

ORCID

Nature Methods is committed to improving transparency in authorship. As part of our efforts in this direction, we are now requesting that all authors identified as 'corresponding author' on published papers create and link their Open Researcher and Contributor Identifier (ORCID) with their account on the Manuscript Tracking System (MTS), prior to acceptance. This applies to primary research papers only. ORCID helps the scientific community achieve unambiguous attribution of all scholarly contributions. You can create and link your ORCID from the home page of the MTS by clicking on 'Modify my Springer Nature account'. For more information please visit <http://www.springernature.com/orcid>.

Sincerely,
Rita

Rita Strack, Ph.D.
Senior Editor
Nature Methods

Reviewers' Comments:

Reviewer #1 (Remarks to the Author):

The submitted manuscript titled "Fourier-Based 3D Multistage Transformer for Aberration Correction in Multicellular Specimens" presents a novel AI-enabled extension of a previous technique, adaptive optics lattice light-sheet microscopy (AO-LLSM), to eliminate the need for the guide star and wavefront sensing hardware and simplifying the experimental workflow. To achieve this, the authors devise a Fourier-based 3D multistage transformer (named AOVIFT), which is applied to estimate the aberration (quantified by Zernike coefficients) directly from the LLSM images of punctate samples. By introducing the Fourier embeddings and elaborately designing the network architecture such as patch sizes and stage numbers, the authors derived the optimized architecture of AOVIFT, which substantially outperforms its counterparts in computational cost, training time, and memory footprint. In live cell imaging experiments, the authors introduce fluorescent puncta of sub-diffractive size by using genome-edited specimens expressing fluorescent protein-fused versions of AP2. This clever and vital approach enables aberration quantification in live specimens and makes AOVIFT available in broad practical AO-LLSM imaging experiments rather than in limited scenarios as some existing methods (e.g., only single molecule localization microscopy for DL-AO [Zhang, P. et al. Nature Methods 20, 1748-1758 (2023)].). Moreover, I particularly like the confidence evaluation method based on digital rotations, which gives users a reliable clue that instruct whether to trust the output of AOVIFT or not.

Generally, the manuscript is well-communicated, excellent, and high precision writing, with beautiful display images. The claims and interpretation of the authors are supported by solid experimental data providing a scientifically sound paper. I'm particularly impressed by the rigorous and comprehensive characterization of the proposed AOVIFT, which covers nearly every aspect that may influence model performance—the network architecture, hyperparameters, training data size, SNR of the data, modes number of Zernike polynomials, etc. And authors comprehensively reported both ability and disability of the method. Although numerous AI-based algorithms are developed in recent years, few of them is able to cooperate well with underlying imaging hardware. I regard AOVIFT an excellent example that synergizes technical advances in both optical hardware and AI algorithms to extend the capability of optical systems. In saying this, I have a few minor comments which it would be good for the authors to address before the paper's acceptance and publication to Nature Methods.

Minor comments:

1. Although there has been a comprehensive ablation study on the choice of baseline models, I didn't find direct comparisons between AOVIFT (e.g., the small version) with and without Fourier embedding input. Since it has been reported that incorporating Fourier embeddings benefit feature extraction of microscopy images in a previous work [Qiao, C. et al. Nature Methods 18, 194-202 (2021).], and the Fourier embedding is also a major selling point of this paper (underlined in the title, abstract, and many other places), the ablation study of Fourier embedding usage and even the normalization step of phase map illustrated in Fig. S4d should be added. This will clarify what proportion of the improvement in efficiency is brought about by the operations in Fourier space.
2. Line 168: the authors mention the introduction of CCP channel "noninvasively produces a robust signal for AO correction that does not preclude simultaneously imaging another subcellular target that occupies the same fluorescence channel". This

claim is confusing for me in that if the CCPs occupy a fluorescence channel, this channel cannot be used to label another type of structure. To this end, does this claim mean that another biological structure that occupies the same channel as the CCPs do not influence the performance of AOVIFT? Which experimental result supports this argument? Or there are some tricks to image two structures independently within a single color channel?

3. In the zebrafish embryo images shown in Fig. 5a, I notice severe background fluorescence beneath the signal from AP2. Does this background influence the aberration prediction of AOVIFT? If yes, is it possible to quantitatively evaluate the influence of background on AOVIFT?

4. Line 366, the authors define the criterion of an image achieving diffraction limited performance when its aberration $<0.075\lambda$ rms. And in line 2015, this is changed into "a quarter of a wave". To my knowledge, the diffraction limit definition in terms of aberration is not a common sense for the community. Thus related references should be cited here.

5. In Fig. S21b,d,f and Fig. S22 b,d,f, I'm confused that why there seems no aberration correction effect for images with aberration amplitudes larger than 0.2λ ?

6. There are some typos that should be correct: Line 410, "100 mm" should be "100 nm"? Line 2111, the period after "(Supplementary Fig. S21e)" should be a comma?

Reviewer #1 (Remarks on code availability):

Due to time constraints I have been unable to test the code but do intend to do so in the coming weeks.

Reviewer #2 (Remarks to the Author):

This manuscript introduces AOVIFT, a novel machine learning-based framework for aberration correction in microscopy. The method utilizes a 3D multistage Vision Transformer operating on Fourier domain embeddings to infer and correct aberrations. The authors demonstrate the efficacy of AOVIFT in correcting aberrations in both synthetic and real biological samples, showcasing its potential to improve high-resolution volumetric microscopy. This is a well-written and significant contribution to the field of microscopy. I recommend accepting this manuscript for publication in Nature Methods after the authors have addressed my concerns suggested below.

1. I am still not entirely clear on the necessity of operating in the Fourier domain. Could the authors please explain this more clearly?

2. While the authors discuss the performance of AOVIFT under various conditions, a more thorough discussion of the limitations of the method would be beneficial. For example, what are the limitations in terms of the density of fluorescent puncta, the magnitude of aberrations, or the speed of specimen motion that AOVIFT can handle?

3. If thicker samples are imaged, it implies that higher-order Zernike modes may become dominant in the aberrations. According to the authors' predictions, would the algorithm still function to some extent in this situation, or would it completely break down?

4. When imaging live specimens using the authors' method, how large can each isoplanatic region be? Can the authors provide any empirical guidelines?

5. When using spatially varying deconvolution, how can one reduce discontinuities between images of adjacent isoplanatic regions, as long as diverse PSFs are applied?

Reviewer #3 (Remarks to the Author):

This paper summarises a new method of essentially using AI to analyze distorted images produced by microscopes and therefore to be able to produce a corrected image. It is a modified form of using adaptive optics.

First I will list what I see as the positive points. It is a very well written and exhaustive manuscript. The figures are all excellent and the results are clear. This is a timely topic and the results will be of broad interest. The results are also certainly impressive and novel.

On the downside, the fundamental technique I would argue is not new. There are lots of papers on adaptive optics in microscopy and there are also a number of papers on using machine learning - as the authors have referenced. So this is essentially a paper on a new form of machine learning applied to adaptive optics in microscopy - which has a number of advantages (detailed in the manuscript). It also has the (small) drawback genome-edited specimens are required.

Pulling the positive and negatives together - I'd say that this is an excellent paper that certainly warrants publication. If the other referees are equally positive then I'd be very happy to see this published in Nature Methods. My only reservation is that I'd say this is on the borderline of being suitable for Nature Methods. It is a variation on an existing technique and I would equally expect to see this in one of the very good optics journals where this type of result is published.

Version 1:

Decision Letter:

Our ref: NMETH-A60229A

26th Jun 2025

Dear Thayer,

Thank you for submitting your revised manuscript "Fourier-Based 3D Multistage Transformer for Aberration Correction in Multicellular Specimens" (NMEMH-A60229A). It has now been seen by the original referees and their comments are below. The reviewers find that the paper has improved in revision, and therefore we'll be happy in principle to publish it in Nature Methods, pending minor revisions to comply with our editorial and formatting guidelines.

TRANSPARENT PEER REVIEW

ORCID

Sincerely,
Rita

Rita Strack, Ph.D.
Senior Editor
Nature Methods

Reviewer #1 (Remarks to the Author):

The authors have satisfactorily addressed my comments and resolved my concerns. In the revision, the authors discussed the effectiveness of Fourier-embedded inputs with three new extended figures, which I find convincing and scientifically sound. Moreover, the authors revised claims that would have been ambiguous as suggested. I am thus supportive of publication in Nature Methods.

Reviewer #1 (Remarks on code availability):

Repo of AOVIFT in GitHub is well organized. Code is clean and straightforward to follow. Detailed and thorough installation instructions.

Reviewer #2 (Remarks to the Author):

I think the authors have fully addressed all my concerns. I have no further comments and would like to support publication of their manuscript.

Reviewer #3 (Remarks to the Author):

I am happy with the changes that the authors have made to all the comments from the referees. My major comment was that I thought this paper was borderline for publication in Nature Methods (despite being an excellent paper) but I can see the other two reviewers were more keen, and I also think the authors have responded to my point.

Version 2:

Decision Letter:

22nd Aug 2025

Dear Thayer,

I am pleased to inform you that your Article, "Fourier-Based 3D Multistage Transformer for Aberration Correction in Multicellular Specimens", has now been accepted for publication in Nature Methods. The received and accepted dates will be March 20, 2025 and August 22, 2025. This note is intended to let you know what to expect from us over the next month or so, and to let you know where to address any further questions.

Over the next few weeks, your paper will be copyedited to ensure that it conforms to Nature Methods style. Once your paper is typeset, you will receive an email with a link to choose the appropriate publishing options for your paper and our Author Services team will be in touch regarding any additional information that may be required. It is extremely important that you let us know now whether you will be difficult to contact over the next month. If this is the case, we ask that you send us the contact information (email, phone and fax) of someone who will be able to check the proofs and deal with any last-minute problems.

Authors may need to take specific actions to achieve compliance with funder and institutional open access mandates.

If your research is supported by a funder that requires immediate open access (e.g. according to [Plan S principles](https://www.springernature.com/gp/open-science/plan-s-compliance) or the [NIH public access policy](https://www.springernature.com/gp/open-science/us-federal-agency-compliance)) then you should select the gold OA route, and we will direct you to the compliant route where possible. Because authors warrant under our subscription licensing terms that they haven't committed to licensing any version of their article under a licence inconsistent with the terms of our agreement – including the applicable embargo period – publication under the subscription model isn't suitable for authors whose funders require no embargo.

If you are active on Twitter/X or Bluesky, please e-mail me your and your coauthors' handles so that we may tag you when the paper is published.

Best regards,
Rita

Rita Strack, Ph.D.
Senior Editor
Nature Methods

Visit the Springer Nature Editorial and Publishing website at http://editorial-jobs.springernature.com?utm_source=ejP_NMeth_email&utm_medium=ejP_NMeth_email&utm_campaign=ejp_Nmeth > www.springernature.com/editorial-and-publishing-jobs for more information about our career opportunities. If you have any questions please click [here](mailto:editorial.publishing.jobs@springernature.com) .**

Open Access This Peer Review File is licensed under a Creative Commons Attribution 4.0 International License, which permits use, sharing, adaptation, distribution and reproduction in any medium or format, as long as you give appropriate credit to the original author(s) and the source, provide a link to the Creative Commons license, and indicate if changes were made. In cases where reviewers are anonymous, credit should be given to 'Anonymous Referee' and the source. The images or other third party material in this Peer Review File are included in the article's Creative Commons license, unless indicated otherwise in a credit line to the material. If material is not included in the article's Creative Commons license and your intended use is not permitted by statutory regulation or exceeds the permitted use, you will need to obtain permission directly from the copyright holder.

Reply to Reviews

Submission: NMETH-A60229

Manuscript: Fourier-Based 3D Multistage Transformer for Aberration Correction in Multicellular Specimens

Comment

Your Article has now been seen by three reviewers. As you will see from their comments below, although the reviewers find your work of considerable potential interest, they have raised a number of concerns. We are interested in the possibility of publishing your paper in Nature Methods, but would like to consider your response to these concerns before we reach a final decision on publication. We therefore invite you to revise your manuscript to address these concerns. In addition to addressing the technical concerns about performance and questions, we ask that you focus your revision on clarifying the sample space in which the methods works well, for the biologist reader.

Dear Dr. Strack,

Thank you for the continued consideration of our manuscript. We thank the reviewers for their feedback. We have revised our manuscript and supplementary materials to address their technical concerns and clarify the applicable sample space of AOVIFT. Major revisions are summarized below:

1. We added a new supplementary section with three new extended data figures (Supplementary Figs. S5-S7) to show the importance of our Fourier embedding and address the reviewers' questions about our Fourier embedding design (Appendix A.1).
2. We revised the main text to better explain our results based on specific reviewer comments.
3. All edits have been highlighted in our **'edits.pdf'**

Reviewer 1

Comment

The submitted manuscript titled “Fourier-Based 3D Multistage Transformer for Aberration Correction in Multicellular Specimens” presents a novel AI-enabled extension of a previous technique, adaptive optics lattice light-sheet microscopy (AO-LLSM), to eliminate the need for the guide star and wavefront sensing hardware and simplifying the experimental workflow. To achieve this, the authors devise a Fourier-based 3D multistage transformer (named AOViFT), which is applied to estimate the aberration (quantified by Zernike coefficients) directly from the LLSM images of punctate samples. By introducing the Fourier embeddings and elaborately designing the network architecture such as patch sizes and stage numbers, the authors derived the optimized architecture of AOViFT, which substantially outperforms its counterparts in computational cost, training time, and memory footprint. In live cell imaging experiments, the authors introduce fluorescent puncta of sub-diffractive size by using genome-edited specimens expressing fluorescent protein-fused versions of AP2. This clever and vital approach enables aberration quantification in live specimens and makes AOViFT available in broad practical AO-LLSM imaging experiments rather than in limited scenarios as some existing methods (e.g., only single molecule localization microscopy for DL-AO [Zhang, P. et al. Nature Methods 20, 1748-1758 (2023)].). Moreover, I particularly like the confidence evaluation method based on digital rotations, which gives users a reliable clue that instruct whether to trust the output of AOViFT or not. Generally, the manuscript is well-communicated, excellent, and high precision writing, with beautiful display images. The claims and interpretation of the authors are supported by solid experimental data providing a scientifically sound paper. I’m particularly impressed by the rigorous and comprehensive characterization of the proposed AOViFT, which covers nearly every aspect that may influence model performance—the network architecture, hyperparameters, training data size, SNR of the data, modes number of Zernike polynomials, etc. And authors comprehensively reported both ability and disability of the method. Although numerous AI-based algorithms are developed in recent years, few of them is able to cooperate well with underlying imaging hardware. I regard AOViFT an excellent example that synergizes technical advances in both optical hardware and AI algorithms to extend the capability of optical systems. In saying this, I have a few minor comments which it would be good for the authors to address before the paper’s acceptance and publication to Nature Methods.

Although there has been a comprehensive ablation study on the choice of baseline models, I didn’t find direct comparisons between AOViFT (e.g., the small version) with and without Fourier embedding input. Since it has been reported that incorporating Fourier embeddings benefit feature extraction of microscopy images in a previous work [Qiao, C. et al. Nature Methods 18, 194-202 (2021).], and the Fourier embedding is also a major selling point of this paper (underlined in the title, abstract, and many other places), the ablation study of Fourier embedding usage and even the normalization step of phase map illustrated in Fig. S4d should be added. This will clarify what proportion of the improvement in efficiency is brought about by the operations in Fourier space.

We trained models on raw (real-space) inputs, but they failed to generalize beyond the training set. Without a 3D foundation model pretrained on an immense microscopy dataset covering the breath of biological features, a broadly applicable real-space solution proved impractical. We have

expanded our discussion in Appendix A.1 and added Supplementary Figs. S5-S7, which compare the generalization performance of Fourier-embedded inputs against three real-space examples.

Comment

Line 168: the authors mention the introduction of CCP channel “noninvasively produces a robust signal for AO correction that does not preclude simultaneously imaging another subcellular target that occupies the same fluorescence channel”. This claim is confusing for me in that if the CCPs occupy a fluorescence channel, this channel cannot be used to label another type of structure. To this end, does this claim mean that another biological structure that occupies the same channel as the CCPs do not influence the performance of AOVIFT? Which experimental result supports this argument? Or there are some tricks to image two structure independently within a single color channel?

We revised this sentence for clarity. Although we have not experimentally tested our model on mixtures of sub-diffractive objects and other subcellular targets, appropriate preprocessing using conventional computer-vision (e.g., peak-local-maxima detection) or emerging deep-learning methods (e.g., MicroSplit¹) could facilitate this approach. Alternatively, a priori knowledge of distinct spatial confinement can help delineate targets within the same channel. For instance, membrane-bound AP2 assemblies could likely be spatially resolved in 3D from non-membrane structures (e.g., nuclei or most mitochondria) imaged together, aiding the isolation of AP2 puncta for our model. In this work, however, we confine each target to its own fluorescence channel to simplify model validation.

Comment

In the zebrafish embryo images shown in Fig. 5a, I notice severe background fluorescence beneath the signal from AP2. Does this background influence the aberration prediction of AOVIFT? If yes, is it possible to quantitatively evaluate the influence of background on AOVIFT?

Evaluating AOVIFT’s background dependence in live samples is challenging, but our photon-vs-aberration study (Figs. 2, S10–S13, S20, S24–S25) clarifies the requirements. The model needs a few hundred photons above the local low-frequency background (e.g., aberration-based blooming, cytosolic fluorescence, compromised light-sheet confinement, sample scattering) to converge in one iteration. Our Fourier-domain filter (see Methods) removes high-frequency noise beyond the OTF support, but any residual background that masks true high-spatial-frequency content will slow convergence, i.e., regions with stronger background simply require more iterations. We also avoid aggressive local background subtraction since it can strip away the subtle, high-frequency details that carry the aberration fingerprint.

¹Ashesh, et al. Microsplit: Semantic unmixing of fluorescent microscopy data. BioRxiv, 2025.

Comment

Line 366, the authors define the criterion of an image achieving diffraction limited performance when its aberration $< 0.075\lambda$ RMS. And in line 2015, this is changed into “a quarter of a wave”. To my knowledge, the diffraction limit definition in terms of aberration is not a common sense for the community. Thus, related references should be cited here.

In results section 2.2, we have added context and citations to justify our $< 0.075\lambda$ RMS criterion and standardized all terminology to RMS. The criterion we chose was based upon the Rayleigh quarter-wave criterion and a Strehl ratio of $0.8^{2,3}$. It allows for a maximum wavefront error of $\lambda/4$, peak-to-valley, below which image quality is “not sensibly degraded”. The peak-to-valley metric is also a familiar metric to readers as it is a common specification on commercial optical surfaces (e.g., a mirror with physical $\lambda/8$ flatness would impart an $\lambda/4$ optical wavefront error⁴). While wavefront peak-to-valley measures the distortion from two points on the surface, root-mean-squared (RMS) gives a statistical measurement of the wavefront, and in practice, both are widely used.

Comment

In Fig. S21b,d,f and Fig. S22 b,d,f, I'm confused that why there seems no aberration correction effect for images with aberration amplitudes larger than 0.2λ ?

We corrected the captions to clarify that panels b, d, and f show the uncorrected MIPs at varying aberration amplitudes and SNR levels.

Comment

There are some typos that should be correct: Line 410, “100 mm” should be “100 nm”? Line 2111, the period after “(Supplementary Fig. S21e)” should be a comma?

Revised. Thank you!

²Bentley, B, and Olson, C. Field Guide to Lens Design. SPIE Press, 2012.

³Mahajan, V N. Strehl ratio for primary aberrations: some analytical results for circular and annular pupils. Journal of the Optical Society of America, 1982.

⁴<https://www.newport.com/n/optical-surfaces>

Reviewer 2

Comment

This manuscript introduces AOVIFT, a novel machine learning-based framework for aberration correction in microscopy. The method utilizes a 3D multistage Vision Transformer operating on Fourier domain embeddings to infer and correct aberrations. The authors demonstrate the efficacy of AOVIFT in correcting aberrations in both synthetic and real biological samples, showcasing its potential to improve high-resolution volumetric microscopy. This is a well-written and significant contribution to the field of microscopy. I recommend accepting this manuscript for publication in Nature Methods after the authors have addressed my concerns suggested below. I am still not entirely clear on the necessity of operating in the Fourier domain. Could the authors please explain this more clearly?

We have added Appendix A.1 to explain why Fourier-domain embeddings are essential, and included Supplementary Figs. S5–S7 to compare the generalization performance of Fourier-embedded inputs against real-space examples.

Comment

While the authors discuss the performance of AOVIFT under various conditions, a more thorough discussion of the limitations of the method would be beneficial. For example, what are the limitations in terms of the density of fluorescent puncta, the magnitude of aberrations, or the speed of specimen motion that AOVIFT can handle?

We have provided detailed breakdown of the model performance under various conditions in Appendix C, including under extreme aberrations over a wide range of SNR levels. We summarize the key points here:

Magnitude of aberrations: The AOVIFT “Small” model can successfully recover large aberrations (up to 0.7λ RMS). For such large aberration magnitudes, achieving diffraction-limited performance requires multiple iterations. Full details are in Appendix C.1, and Supplementary Fig. S10.

Density of fluorescent puncta: We evaluated our model performance across a range of puncta densities (Appendix C.3). Our models, even when trained on up to five synthetic beads in a 64^3 FOV, showed no degradation in their performance when tested on FOVs with up to 150 puncta, provided their mean nearest neighbor distance is $> 400\text{nm}$ (Supplementary Fig. S13).

Speed of specimen motion: Directly quantifying and correcting for motion is challenging due to multiple sources contributing at various timescales (e.g., heartbeat, cell migration, subcellular dynamics). AOVIFT assumes a quasi-static PSF for each analyzed tile. Consequently, PSF changes during acquisition leading to blurred, enlarged, or lower SNR of the puncta will obscure the aberration fingerprint in the Fourier embeddings and degrade model performance. We evaluated our model’s sensitivity to empirically estimated motion blurs, which causes puncta to appear larger, in

Appendix C.4 and Supplementary Fig. S26.

Comment

If thicker samples are imaged, it implies that higher-order Zernike modes may become dominant in the aberrations. According to the authors' predictions, would the algorithm still function to some extent in this situation, or would it completely break down?

While thicker samples can introduce more complex aberrations, the observable dominance and correction capability of these higher-order modes are influenced by detection NA and deformable mirror limitations. AOVIFT is pretrained to correct the first 15 modes, and aligned with our current range of Zernike mode sensitivity and correctability. AOVIFT is designed for flexibility, and may be retrained using the data simulation framework provided to predict higher-order Zernike modes.

Comment

When imaging live specimens using the authors' method, how large can each isoplanatic region be? Can the authors provide any empirical guidelines?

The size of the isoplanatic patch is highly variable depending primarily on tissue heterogeneity and imaging depth. For example, the isoplanatic patch near the surface or the zebrafish tail can be relatively homogeneous and span several tens of microns. However, highly heterogeneous areas or those with strong intrinsic lensing effects, such as the notochord, exhibit much smaller isoplanatic patches (3–10 μm^3). A key utility of AOVIFT is precisely its ability to empirically map these varying isoplanatic patches across a large field of view. Fig. 6 demonstrates this by showing an isoplanatic patch map near the notochord. Despite similar broader astigmatic aberrations across the FOV, a wide range of spatially varying aberrations is evident between neighboring tiles.

To provide further empirical context, our previous work imaging developing zebrafish embryos with adaptive optical lattice light sheet microscopy⁵ showed that for more homogeneous tissues without sharp internal transitions, an average aberration correction could be effectively applied over an imaging volume of approximately $30 \times 30 \times 30 \mu\text{m}^3$. This earlier finding underscores that larger regions of isoplanaticity can exist in less complex tissue, and conversely, that sharp transitions between tissue types significantly reduce the size of the isoplanatic patch.

Comment

When using spatially varying deconvolution, how can one reduce discontinuities between images of adjacent isoplanatic regions, as long as diverse PSFs are applied?

⁵Liu, T, et al. Observing the cell in its native state: Imaging subcellular dynamics in multicellular organisms. Science (2018).

We detailed our method for minimizing boundary discontinuities in Methods section 4.5, which involves tiling the image on a regular grid. We extended each processing tile by half the PSF window width (32 pixels here) at each boundary, discarding these extensions after deconvolution. This significantly reduces border artifacts by ensuring stitched core regions are processed with more complete boundary information.

To further mitigate discontinuities, especially when PSFs vary significantly between regions, several additional strategies could be employed:

Optimized tile sizing: Where signal density and resolution permit local PSF prediction, reducing tile size can promote more gradual PSF variations between adjacent tiles, thereby decreasing discontinuities.

Contextual PSF estimation: For low-signal regions where direct PSF reconstruction is difficult, interpolating a local PSF from the wavefronts of well-characterized neighboring tiles, rather than using an ideal PSF, ensures smoother PSF transitions.

Adaptive region segmentation: For areas with sharp aberration changes not well captured by a regular grid (e.g., due to sparse puncta for PSF estimation), an advanced ROI generator could define irregular regions that better match true isoplanatic patches. Applying a specific PSF to such adaptively segmented regions up to their natural boundaries would reduce discontinuities by aligning the PSF changes more closely with the sample's optical transitions.

Reviewer 3

Comment

This paper summarises a new method of essentially using AI to analyze distorted images produced by microscopes and therefore to be able to produce a corrected image. It is a modified form of using adaptive optics. First I will list what I see as the positive points. It is a very well written and exhaustive manuscript. The figures are all excellent and the results are clear. This is a timely topic and the results will be of broad interest. The results are also certainly impressive and novel. On the downside, the fundamental technique I would argue is not new. There are lots of papers on adaptive optics in microscopy and there are also a number of papers on using machine learning - as the authors have referenced. So this is essentially a paper on a new form of machine learning applied to adaptive optics in microscopy - which has a number of advantages (detailed in the manuscript). It also has the (small) draw back genome-edited specimens are required. Pulling the positive and negatives together - I'd say that this is an excellent paper that certainly warrants publication. If the other referees are equally positive then I'd be very happy to see this published in Nature Methods. My only reservation is that I'd say this is on the borderline of being suitable for Nature Methods. It is a variation on an existing technique and I would equally expect to see this in one of the very good optics journals where this type of result is published.

While the general concept of using AI for adaptive optics has been explored, AOViT is specifically designed to overcome key limitations of prior methods, particularly for live, volumetric imaging like adaptive optical lattice light sheet microscopy.

AOViT represents a new tool, complementary to traditional hardware-based methods, capable of significantly lowering technical barriers and enabling the study of complex biological processes in their native context with unprecedented clarity and scale. AOViT operates directly on Fourier domain embeddings, a fundamentally different input data representation. This approach is motivated by the insight that aberrations globally fingerprint the Fourier transform. AOViT's 3D multistage backbone also offers a unique architecture for learning features across different scales while maintaining accurate prediction and better utilization of the compute resources than existing approaches.

The development of this AI framework, along with its specialized synthetic data pipelines, presents a practical solution tailored for high-resolution live imaging.

Reply to Reviews

Submission: NMETH-A60229

Manuscript: Fourier-Based 3D Multistage Transformer for Aberration Correction in Multicellular Specimens

Comment

Your Article has now been seen by three reviewers. As you will see from their comments below, although the reviewers find your work of considerable potential interest, they have raised a number of concerns. We are interested in the possibility of publishing your paper in Nature Methods, but would like to consider your response to these concerns before we reach a final decision on publication. We therefore invite you to revise your manuscript to address these concerns. In addition to addressing the technical concerns about performance and questions, we ask that you focus your revision on clarifying the sample space in which the methods works well, for the biologist reader.

Dear Dr. Strack,

Thank you for the continued consideration of our manuscript. We thank the reviewers for their feedback. We have revised our manuscript and supplementary materials to address their technical concerns and clarify the applicable sample space of AOVIFT. Major revisions are summarized below:

1. We added a new supplementary section with three new extended data figures (Supplementary Figs. S5-S7) to show the importance of our Fourier embedding and address the reviewers' questions about our Fourier embedding design (Appendix A.1).
2. We revised the main text to better explain our results based on specific reviewer comments.
3. All edits have been highlighted in our **'edits.pdf'**

Reviewer 1

Comment

The submitted manuscript titled “Fourier-Based 3D Multistage Transformer for Aberration Correction in Multicellular Specimens” presents a novel AI-enabled extension of a previous technique, adaptive optics lattice light-sheet microscopy (AO-LLSM), to eliminate the need for the guide star and wavefront sensing hardware and simplifying the experimental workflow. To achieve this, the authors devise a Fourier-based 3D multistage transformer (named AOViFT), which is applied to estimate the aberration (quantified by Zernike coefficients) directly from the LLSM images of punctate samples. By introducing the Fourier embeddings and elaborately designing the network architecture such as patch sizes and stage numbers, the authors derived the optimized architecture of AOViFT, which substantially outperforms its counterparts in computational cost, training time, and memory footprint. In live cell imaging experiments, the authors introduce fluorescent puncta of sub-diffractive size by using genome-edited specimens expressing fluorescent protein-fused versions of AP2. This clever and vital approach enables aberration quantification in live specimens and makes AOViFT available in broad practical AO-LLSM imaging experiments rather than in limited scenarios as some existing methods (e.g., only single molecule localization microscopy for DL-AO [Zhang, P. et al. Nature Methods 20, 1748-1758 (2023)].). Moreover, I particularly like the confidence evaluation method based on digital rotations, which gives users a reliable clue that instruct whether to trust the output of AOViFT or not. Generally, the manuscript is well-communicated, excellent, and high precision writing, with beautiful display images. The claims and interpretation of the authors are supported by solid experimental data providing a scientifically sound paper. I’m particularly impressed by the rigorous and comprehensive characterization of the proposed AOViFT, which covers nearly every aspect that may influence model performance—the network architecture, hyperparameters, training data size, SNR of the data, modes number of Zernike polynomials, etc. And authors comprehensively reported both ability and disability of the method. Although numerous AI-based algorithms are developed in recent years, few of them is able to cooperate well with underlying imaging hardware. I regard AOViFT an excellent example that synergizes technical advances in both optical hardware and AI algorithms to extend the capability of optical systems. In saying this, I have a few minor comments which it would be good for the authors to address before the paper’s acceptance and publication to Nature Methods.

Although there has been a comprehensive ablation study on the choice of baseline models, I didn’t find direct comparisons between AOViFT (e.g., the small version) with and without Fourier embedding input. Since it has been reported that incorporating Fourier embeddings benefit feature extraction of microscopy images in a previous work [Qiao, C. et al. Nature Methods 18, 194-202 (2021).], and the Fourier embedding is also a major selling point of this paper (underlined in the title, abstract, and many other places), the ablation study of Fourier embedding usage and even the normalization step of phase map illustrated in Fig. S4d should be added. This will clarify what proportion of the improvement in efficiency is brought about by the operations in Fourier space.

We trained models on raw (real-space) inputs, but they failed to generalize beyond the training set. Without a 3D foundation model pretrained on an immense microscopy dataset covering the breath of biological features, a broadly applicable real-space solution proved impractical. We have

expanded our discussion in Appendix A.1 and added Supplementary Figs. S5-S7, which compare the generalization performance of Fourier-embedded inputs against three real-space examples.

Comment

Line 168: the authors mention the introduction of CCP channel “noninvasively produces a robust signal for AO correction that does not preclude simultaneously imaging another subcellular target that occupies the same fluorescence channel”. This claim is confusing for me in that if the CCPs occupy a fluorescence channel, this channel cannot be used to label another type of structure. To this end, does this claim mean that another biological structure that occupies the same channel as the CCPs do not influence the performance of AOVIFT? Which experimental result supports this argument? Or there are some tricks to image two structure independently within a single color channel?

We revised this sentence for clarity. Although we have not experimentally tested our model on mixtures of sub-diffractive objects and other subcellular targets, appropriate preprocessing using conventional computer-vision (e.g., peak-local-maxima detection) or emerging deep-learning methods (e.g., MicroSplit¹) could facilitate this approach. Alternatively, a priori knowledge of distinct spatial confinement can help delineate targets within the same channel. For instance, membrane-bound AP2 assemblies could likely be spatially resolved in 3D from non-membrane structures (e.g., nuclei or most mitochondria) imaged together, aiding the isolation of AP2 puncta for our model. In this work, however, we confine each target to its own fluorescence channel to simplify model validation.

Comment

In the zebrafish embryo images shown in Fig. 5a, I notice severe background fluorescence beneath the signal from AP2. Does this background influence the aberration prediction of AOVIFT? If yes, is it possible to quantitatively evaluate the influence of background on AOVIFT?

Evaluating AOVIFT’s background dependence in live samples is challenging, but our photon-vs-aberration study (Figs. 2, S10–S13, S20, S24–S25) clarifies the requirements. The model needs a few hundred photons above the local low-frequency background (e.g., aberration-based blooming, cytosolic fluorescence, compromised light-sheet confinement, sample scattering) to converge in one iteration. Our Fourier-domain filter (see Methods) removes high-frequency noise beyond the OTF support, but any residual background that masks true high-spatial-frequency content will slow convergence, i.e., regions with stronger background simply require more iterations. We also avoid aggressive local background subtraction since it can strip away the subtle, high-frequency details that carry the aberration fingerprint.

¹Ashesh, et al. Microsplit: Semantic unmixing of fluorescent microscopy data. BioRxiv, 2025.

Comment

Line 366, the authors define the criterion of an image achieving diffraction limited performance when its aberration $< 0.075\lambda$ RMS. And in line 2015, this is changed into “a quarter of a wave”. To my knowledge, the diffraction limit definition in terms of aberration is not a common sense for the community. Thus, related references should be cited here.

In results section 2.2, we have added context and citations to justify our $< 0.075\lambda$ RMS criterion and standardized all terminology to RMS. The criterion we chose was based upon the Rayleigh quarter-wave criterion and a Strehl ratio of $0.8^{2,3}$. It allows for a maximum wavefront error of $\lambda/4$, peak-to-valley, below which image quality is “not sensibly degraded”. The peak-to-valley metric is also a familiar metric to readers as it is a common specification on commercial optical surfaces (e.g., a mirror with physical $\lambda/8$ flatness would impart an $\lambda/4$ optical wavefront error⁴). While wavefront peak-to-valley measures the distortion from two points on the surface, root-mean-squared (RMS) gives a statistical measurement of the wavefront, and in practice, both are widely used.

Comment

In Fig. S21b,d,f and Fig. S22 b,d,f, I'm confused that why there seems no aberration correction effect for images with aberration amplitudes larger than 0.2λ ?

We corrected the captions to clarify that panels b, d, and f show the uncorrected MIPs at varying aberration amplitudes and SNR levels.

Comment

There are some typos that should be correct: Line 410, “100 mm” should be “100 nm”? Line 2111, the period after “(Supplementary Fig. S21e)” should be a comma?

Revised. Thank you!

²Bentley, B, and Olson, C. Field Guide to Lens Design. SPIE Press, 2012.

³Mahajan, V N. Strehl ratio for primary aberrations: some analytical results for circular and annular pupils. Journal of the Optical Society of America, 1982.

⁴<https://www.newport.com/n/optical-surfaces>

Reviewer 2

Comment

This manuscript introduces AOVIFT, a novel machine learning-based framework for aberration correction in microscopy. The method utilizes a 3D multistage Vision Transformer operating on Fourier domain embeddings to infer and correct aberrations. The authors demonstrate the efficacy of AOVIFT in correcting aberrations in both synthetic and real biological samples, showcasing its potential to improve high-resolution volumetric microscopy. This is a well-written and significant contribution to the field of microscopy. I recommend accepting this manuscript for publication in Nature Methods after the authors have addressed my concerns suggested below. I am still not entirely clear on the necessity of operating in the Fourier domain. Could the authors please explain this more clearly?

We have added Appendix A.1 to explain why Fourier-domain embeddings are essential, and included Supplementary Figs. S5–S7 to compare the generalization performance of Fourier-embedded inputs against real-space examples.

Comment

While the authors discuss the performance of AOVIFT under various conditions, a more thorough discussion of the limitations of the method would be beneficial. For example, what are the limitations in terms of the density of fluorescent puncta, the magnitude of aberrations, or the speed of specimen motion that AOVIFT can handle?

We have provided detailed breakdown of the model performance under various conditions in Appendix C, including under extreme aberrations over a wide range of SNR levels. We summarize the key points here:

Magnitude of aberrations: The AOVIFT “Small” model can successfully recover large aberrations (up to 0.7λ RMS). For such large aberration magnitudes, achieving diffraction-limited performance requires multiple iterations. Full details are in Appendix C.1, and Supplementary Fig. S10.

Density of fluorescent puncta: We evaluated our model performance across a range of puncta densities (Appendix C.3). Our models, even when trained on up to five synthetic beads in a 64^3 FOV, showed no degradation in their performance when tested on FOVs with up to 150 puncta, provided their mean nearest neighbor distance is $> 400\text{nm}$ (Supplementary Fig. S13).

Speed of specimen motion: Directly quantifying and correcting for motion is challenging due to multiple sources contributing at various timescales (e.g., heartbeat, cell migration, subcellular dynamics). AOVIFT assumes a quasi-static PSF for each analyzed tile. Consequently, PSF changes during acquisition leading to blurred, enlarged, or lower SNR of the puncta will obscure the aberration fingerprint in the Fourier embeddings and degrade model performance. We evaluated our model’s sensitivity to empirically estimated motion blurs, which causes puncta to appear larger, in

Appendix C.4 and Supplementary Fig. S26.

Comment

If thicker samples are imaged, it implies that higher-order Zernike modes may become dominant in the aberrations. According to the authors' predictions, would the algorithm still function to some extent in this situation, or would it completely break down?

While thicker samples can introduce more complex aberrations, the observable dominance and correction capability of these higher-order modes are influenced by detection NA and deformable mirror limitations. AOVIFT is pretrained to correct the first 15 modes, and aligned with our current range of Zernike mode sensitivity and correctability. AOVIFT is designed for flexibility, and may be retrained using the data simulation framework provided to predict higher-order Zernike modes.

Comment

When imaging live specimens using the authors' method, how large can each isoplanatic region be? Can the authors provide any empirical guidelines?

The size of the isoplanatic patch is highly variable depending primarily on tissue heterogeneity and imaging depth. For example, the isoplanatic patch near the surface or the zebrafish tail can be relatively homogeneous and span several tens of microns. However, highly heterogeneous areas or those with strong intrinsic lensing effects, such as the notochord, exhibit much smaller isoplanatic patches (3–10 μm^3). A key utility of AOVIFT is precisely its ability to empirically map these varying isoplanatic patches across a large field of view. Fig. 6 demonstrates this by showing an isoplanatic patch map near the notochord. Despite similar broader astigmatic aberrations across the FOV, a wide range of spatially varying aberrations is evident between neighboring tiles.

To provide further empirical context, our previous work imaging developing zebrafish embryos with adaptive optical lattice light sheet microscopy⁵ showed that for more homogeneous tissues without sharp internal transitions, an average aberration correction could be effectively applied over an imaging volume of approximately $30 \times 30 \times 30 \mu\text{m}^3$. This earlier finding underscores that larger regions of isoplanaticity can exist in less complex tissue, and conversely, that sharp transitions between tissue types significantly reduce the size of the isoplanatic patch.

Comment

When using spatially varying deconvolution, how can one reduce discontinuities between images of adjacent isoplanatic regions, as long as diverse PSFs are applied?

⁵Liu, T, et al. Observing the cell in its native state: Imaging subcellular dynamics in multicellular organisms. Science (2018).

We detailed our method for minimizing boundary discontinuities in Methods section 4.5, which involves tiling the image on a regular grid. We extended each processing tile by half the PSF window width (32 pixels here) at each boundary, discarding these extensions after deconvolution. This significantly reduces border artifacts by ensuring stitched core regions are processed with more complete boundary information.

To further mitigate discontinuities, especially when PSFs vary significantly between regions, several additional strategies could be employed:

Optimized tile sizing: Where signal density and resolution permit local PSF prediction, reducing tile size can promote more gradual PSF variations between adjacent tiles, thereby decreasing discontinuities.

Contextual PSF estimation: For low-signal regions where direct PSF reconstruction is difficult, interpolating a local PSF from the wavefronts of well-characterized neighboring tiles, rather than using an ideal PSF, ensures smoother PSF transitions.

Adaptive region segmentation: For areas with sharp aberration changes not well captured by a regular grid (e.g., due to sparse puncta for PSF estimation), an advanced ROI generator could define irregular regions that better match true isoplanatic patches. Applying a specific PSF to such adaptively segmented regions up to their natural boundaries would reduce discontinuities by aligning the PSF changes more closely with the sample's optical transitions.

Reviewer 3

Comment

This paper summarises a new method of essentially using AI to analyze distorted images produced by microscopes and therefore to be able to produce a corrected image. It is a modified form of using adaptive optics. First I will list what I see as the positive points. It is a very well written and exhaustive manuscript. The figures are all excellent and the results are clear. This is a timely topic and the results will be of broad interest. The results are also certainly impressive and novel. On the downside, the fundamental technique I would argue is not new. There are lots of papers on adaptive optics in microscopy and there are also a number of papers on using machine learning - as the authors have referenced. So this is essentially a paper on a new form of machine learning applied to adaptive optics in microscopy - which has a number of advantages (detailed in the manuscript). It also has the (small) draw back genome-edited specimens are required. Pulling the positive and negatives together - I'd say that this is an excellent paper that certainly warrants publication. If the other referees are equally positive then I'd be very happy to see this published in Nature Methods. My only reservation is that I'd say this is on the borderline of being suitable for Nature Methods. It is a variation on an existing technique and I would equally expect to see this in one of the very good optics journals where this type of result is published.

While the general concept of using AI for adaptive optics has been explored, AOViT is specifically designed to overcome key limitations of prior methods, particularly for live, volumetric imaging like adaptive optical lattice light sheet microscopy.

AOViT represents a new tool, complementary to traditional hardware-based methods, capable of significantly lowering technical barriers and enabling the study of complex biological processes in their native context with unprecedented clarity and scale. AOViT operates directly on Fourier domain embeddings, a fundamentally different input data representation. This approach is motivated by the insight that aberrations globally fingerprint the Fourier transform. AOViT's 3D multistage backbone also offers a unique architecture for learning features across different scales while maintaining accurate prediction and better utilization of the compute resources than existing approaches.

The development of this AI framework, along with its specialized synthetic data pipelines, presents a practical solution tailored for high-resolution live imaging.